# Greedy Feature Construction

**Dino Oglic**[†‡]
dino.oglic@uni-bonn.de
[†]Institut für Informatik III
Universität Bonn, Germany

**Thomas Gärtner**[‡]
thomas.gaertner@nottingham.ac.uk
[‡]School of Computer Science
The University of Nottingham, UK

## Abstract

We present an effective method for supervised feature construction. The main goal of the approach is to construct a feature representation for which a set of linear hypotheses is of sufficient capacity – large enough to contain a satisfactory solution to the considered problem and small enough to allow good generalization from a small number of training examples. We achieve this goal with a greedy procedure that constructs features by empirically fitting squared error residuals. The proposed constructive procedure is consistent and can output a rich set of features. The effectiveness of the approach is evaluated empirically by fitting a linear ridge regression model in the constructed feature space and our empirical results indicate a superior performance of our approach over competing methods.

## 1 Introduction

Every supervised learning algorithm with the ability to generalize from training examples to unseen data points has some type of inductive bias [5]. The bias can be defined as a set of assumptions that together with the training data explain the predictions at unseen points [25]. In order to simplify theoretical analysis of learning algorithms, the inductive bias is often represented by a choice of a hypothesis space (e.g., the inductive bias of linear regression models is the assumption that the relationship between inputs and outputs is linear). The fundamental limitation of learning procedures with an a priori specified hypothesis space (e.g., linear models or kernel methods with a preselected kernel) is that they can learn good concept descriptions only if the hypothesis space selected beforehand is large enough to contain a good solution to the considered problem and small enough to allow good generalization from a small number of training examples. As finding a good hypothesis space is equivalent to finding a good set of features [5], we propose an effective supervised feature construction method to tackle this problem. The main goal of the approach is to embed the data into a feature space for which a set of linear hypotheses is of sufficient capacity. The motivation for this choice of hypotheses is in the desire to exploit the scalability of existing algorithms for training linear models. It is for their scalability that these models are frequently a method of choice for learning on large scale data sets (e.g., the implementation of linear SVM [13] has won the large scale learning challenge at ICML 2008 and KDD CUP 2010). However, as the set of linear hypotheses defined on a small or moderate number of input features is usually of low capacity these methods often learn inaccurate descriptions of target concepts. The proposed approach surmounts this and exploits the scalability of existing algorithms for training linear models while overcoming their low capacity on input features. The latter is achieved by harnessing the information contained in the labeled training data and constructing features by empirically fitting squared error residuals.

We draw motivation for our approach by considering the minimization of the expected squared error using functional gradient descent (Section 2.1). In each step of the descent, the current estimator is updated by moving in the direction of the residual function. We want to mimic this behavior by constructing a feature representation incrementally so that for each step of the descent we add a feature which approximates well the residual function. In this constructive process, we select our features from a predetermined set of basis functions which can be chosen so that a high capacity set

of linear hypotheses corresponds to the constructed feature space (Section 2.2). In our theoretical analysis of the approach, we provide a convergence rate for this constructive procedure (Section 2.3) and give a generalization bound for the empirical fitting of residuals (Section 2.4). The latter is needed because the feature construction is performed based on an independent and identically distributed sample of labeled examples. The approach, presented in Section 2.5, is highly flexible and allows for an extension of a feature representation without complete re-training of the model. As it performs similar to gradient descent, a stopping criteria based on an accuracy threshold can be devised and the algorithm can then be simulated without specifying the number of features *a priori*. In this way, the algorithm can terminate sooner than alternative approaches for simple hypotheses. The method is easy to implement and can be scaled to millions of instances with a parallel implementation.

To evaluate the effectiveness of our approach empirically, we compare it to other related approaches by training linear ridge regression models in the feature spaces constructed by these methods. Our empirical results indicate a superior performance of the proposed approach over competing methods. The results are presented in Section 3 and the approaches are discussed in Section 4.

## 2    Greedy feature construction

In this section, we present our feature construction approach. We start with an overview where we introduce the problem setting and motivate our approach by considering the minimization of the expected squared error using functional gradient descent. Following this, we define a set of features and demonstrate that the approach can construct a rich set of hypotheses. We then show that our greedy constructive procedure converges and give a generalization bound for the empirical fitting of residuals. The section concludes with a pseudo-code description of our approach.

### 2.1    Overview

We consider a learning problem with the squared error loss function where the goal is to find a mapping from a Euclidean space to the set of reals. In these problems, it is typically assumed that a sample $\mathbf{z} = ((x_1, y_1), \dots, (x_m, y_m))$ of $m$ examples is drawn independently from a Borel probability measure $\rho$ defined on $Z = X \times Y$, where $X$ is a compact subset of a finite dimensional Euclidean space with the dot product $\langle \cdot, \cdot \rangle$ and $Y \subset \mathbb{R}$. For every $x \in X$ let $\rho(y \mid x)$ be the conditional probability measure on $Y$ and $\rho_X$ be the marginal probability measure on $X$. For the sake of brevity, when it is clear from the context, we will write $\rho$ instead of $\rho_X$. Let $f_\rho(x) = \int y \, d\rho(y \mid x)$ be the bounded target/regression function of the measure $\rho$. Our goal is to construct a feature representation such that there exists a linear hypothesis on this feature space that approximates well the target function. For an estimator $f$ of the function $f_\rho$ we measure the goodness of fit with the expected squared error in $\rho$, $\mathcal{E}_\rho(f) = \int (f(x) - y)^2 \, d\rho$. The empirical counterpart of the error, defined over a sample $\mathbf{z} \in Z^m$, is denoted with $\mathcal{E}_{\mathbf{z}}(f) = \frac{1}{m} \sum_{i=1}^m (f(x_i) - y_i)^2$.

Having defined the problem setting, we proceed to motivate our approach by considering the minimization of the expected squared error using functional gradient descent. For that, we first review the definition of functional gradient. For a functional $F$ defined on a normed linear space and an element $p$ from this space, the functional gradient $\nabla F(p)$ is the principal linear part of a change in $F$ after it is perturbed in the direction of $q$, $F(p + q) = F(p) + \psi(q) + \epsilon \|q\|$, where $\psi(q)$ is the linear functional with $\nabla F(p)$ as its principal linear part, and $\epsilon \to 0$ as $\|q\| \to 0$ [e.g., see Section 3.2 in 16]. In our case, the normed space is the Hilbert space of square integrable functions $\mathcal{L}_\rho^2(X)$ and for the expected squared error functional on this space we have that it holds

$$\mathcal{E}_\rho(f + \epsilon q) - \mathcal{E}_\rho(f) = \langle 2(f - f_\rho), \epsilon q \rangle_{\mathcal{L}_\rho^2(X)} + \mathcal{O}(\epsilon^2).$$

Hence, an algorithm for the minimization of the expected squared error using functional gradient descent on this space could be specified as

$$f_{t+1} = \nu f_t + 2(1 - \nu)(f_\rho - f_t),$$

where $0 \leq \nu \leq 1$ denotes the learning rate and $f_t$ is the estimate at step $t$. The functional gradient direction $2(f_\rho - f_t)$ is the residual function at step $t$ and the main idea behind our approach is to iteratively refine our feature representation by extending it with a new feature that matches the current residual function. In this way, for a suitable choice of learning rate $\nu$, the functional descent would be performed through a convex hull of features and in each step we would have an estimate of the target function $f_\rho$ expressed as a convex combination of the constructed features.

## 2.2 Greedy features

We introduce now a set of features parameterized with a ridge basis function and hyperparameters controlling the smoothness of these features. As each subset of features corresponds to a set of hypotheses, in this way we specify a family of possible hypothesis spaces. For a particular choice of ridge basis function we argue below that the approach outlined in the previous section can construct a highly expressive feature representation (i.e., a hypothesis space of high capacity).

Let $\mathcal{C}(X)$ be the Banach space of continuous functions on $X$ with the uniform norm. For a Lipschitz continuous function $\phi : \mathbb{R} \to \mathbb{R}$, $\|\phi\|_\infty \leq 1$, and constants $r, s, t > 0$ let $\mathcal{F}_\Theta \subset \mathcal{C}(X)$, $\Theta = (\phi, r, s, t)$, be a set of ridge-wave functions defined on the set $X$,

$$\mathcal{F}_\Theta = \left\{ a\,\phi\left(\langle w, x \rangle + b\right) \mid w \in \mathbb{R}^d, a, b \in \mathbb{R}, |a| \leq r, \|w\|_2 \leq s, |b| \leq t \right\}.$$

From this definition, it follows that for all $g \in \mathcal{F}_\Theta$ it holds $\|g\|_\infty \leq r$. As a ridge-wave function $g \in \mathcal{F}_\Theta$ is bounded and Lipschitz continuous, it is also square integrable in the measure $\rho$ and $g \in \mathcal{L}_\rho^2(X)$. Therefore, $\mathcal{F}_\Theta$ is a subset of the Hilbert space of square integrable functions defined on $X$ with respect to the probability measure $\rho$, i.e., $\mathcal{F}_\Theta \subset \mathcal{L}_\rho^2(X)$.

Taking $\phi(\cdot) = \cos(\cdot)$ in the definition of $\mathcal{F}_\Theta$ we obtain a set of cosine-wave features

$$\mathcal{F}_{\cos} = \left\{ a\cos\left(\langle w, x \rangle + b\right) \mid w \in \mathbb{R}^d, a, b \in \mathbb{R}, |a| \leq r, \|w\|_2 \leq s, |b| \leq t \right\}.$$

For this set of features the approach outlined in Section 2.1 can construct a rich set of hypotheses. To demonstrate this we make a connection to shift-invariant reproducing kernel Hilbert spaces and show that the approach can approximate any bounded function from any shift-invariant reproducing kernel Hilbert space. This means that a set of linear hypotheses defined by cosine features can be of high capacity and our approach can overcome the problems with the low capacity of linear hypotheses defined on few input features. A proof of the following theorem is provided in Appendix B.

**Theorem 1.** *Let $\mathcal{H}_k$ be a reproducing kernel Hilbert space corresponding to a continuous shift-invariant and positive definite kernel $k$ defined on a compact set $X$. Let $\mu$ be the positive and bounded spectral measure whose Fourier transform is the kernel $k$. For any probability measure $\rho$ defined on $X$, it is possible to approximate any bounded function $f \in \mathcal{H}_k$ using a convex combination of $n$ ridge-wave functions from $\mathcal{F}_{\cos}$ such that the approximation error in $\|\cdot\|_\rho$ decays with rate $\mathcal{O}\left(1/\sqrt{n}\right)$.*

## 2.3 Convergence

For the purpose of this paper, it suffices to show the convergence of $\epsilon$-greedy sequences of functions (see Definition 1) in Hilbert spaces. We, however, choose to provide a stronger result which holds for $\epsilon$-greedy sequences in uniformly smooth Banach spaces. In the remainder of the paper, $\mathrm{co}(S)$ and $\overline{S}$ will be used to denote the convex hull of elements from a set $S$ and the closure of $S$, respectively.

**Definition 1.** *Let $\mathcal{B}$ be a Banach space with norm $\|\cdot\|$ and let $S \subseteq \mathcal{B}$. An incremental sequence is any sequence $\{f_n\}_{n \geq 1}$ of elements of $\mathcal{B}$ such that $f_1 \in S$ and for each $n \geq 1$ there is some $g \in S$ so that $f_{n+1} \in \mathrm{co}\left(\{f_n, g\}\right)$. An incremental sequence is greedy with respect to an element $f \in \overline{\mathrm{co}(S)}$ if for all $n \in \mathbb{N}$ it holds $\|f_{n+1} - f\| = \inf\{\|h - f\| \mid h \in \mathrm{co}\left(\{f_n, g\}\right), g \in S\}$. Given a positive sequence of allowed slack terms $\{\epsilon_n\}_{n \geq 1}$, an incremental sequence $\{f_n\}_{n \geq 1}$ is called $\epsilon$-greedy with respect to $f$ if for all $n \in \mathbb{N}$ it holds $\|f_{n+1} - f\| < \inf\{\|h - f\| \mid h \in \mathrm{co}\left(\{f_n, g\}\right), g \in S\} + \epsilon_n$.*

Having introduced the notion of an $\epsilon$-greedy incremental sequence of functions, let us now relate it to our feature construction approach. In the outlined constructive procedure (Section 2.1), we proposed to select new features corresponding to the functional gradient at the current estimate of the target function. Now, if at each step of the functional gradient descent there exists a ridge-wave function from our set of features which approximates well the residual function (w.r.t. $f_\rho$) then this sequence of functions defines a descent through $\mathrm{co}(\mathcal{F}_\Theta)$ which is an $\epsilon$-greedy incremental sequence of functions with respect to $f_\rho \in \overline{\mathrm{co}(\mathcal{F}_\Theta)}$. In Section 2.1, we have also demonstrated that $\mathcal{F}_\Theta$ is a subset of the Hilbert space $\mathcal{L}_\rho^2(X)$ and this is by definition a Banach space.

In accordance with Definition 1, we now consider under what conditions an $\epsilon$-greedy sequence of functions from this space converges to any target function $f_\rho \in \overline{\mathrm{co}(\mathcal{F}_\Theta)}$. Note that this relates to Theorem 1 which confirms the strength of the result by showing that the capacity of $\overline{\mathrm{co}(\mathcal{F}_\Theta)}$ is large. Before we show the convergence of our constructive procedure, we need to prove that an $\epsilon$-greedy

incremental sequence of functions/features can be constructed in our setting. For that, we characterize the Banach spaces in which it is always possible to construct such sequences of functions/features.

**Definition 2.** *Let $\mathcal{B}$ be a Banach space, $\mathcal{B}^*$ the dual space of $\mathcal{B}$, and $f \in \mathcal{B}$, $f \neq 0$. A peak functional for $f$ is a bounded linear operator $F \in \mathcal{B}^*$ such that $\|F\|_{\mathcal{B}^*} = 1$ and $F(f) = \|f\|_{\mathcal{B}}$. The Banach space $\mathcal{B}$ is said to be smooth if for each $f \in \mathcal{B}$, $f \neq 0$, there is a unique peak functional.*

The existence of at least one peak functional for all $f \in \mathcal{B}$, $f \neq 0$, is guaranteed by the Hahn-Banach theorem [27]. For a Hilbert space $\mathcal{H}$, for each element $f \in \mathcal{H}$, $f \neq 0$, there exists a unique peak functional $F = \langle f, \cdot \rangle_{\mathcal{H}} / \|f\|_{\mathcal{H}}$. Thus, every Hilbert space is a smooth Banach space. Donahue et al. [12, Theorem 3.1] have shown that in smooth Banach spaces – and in particular in the Hilbert space $\mathcal{L}^2_\rho(X)$ – an $\epsilon$-greedy incremental sequence of functions can always be constructed. However, not every such sequence of functions converges to the function with respect to which it was constructed. For the convergence to hold, a stronger notion of smoothness is needed.

**Definition 3.** *The modulus of smoothness of a Banach space $\mathcal{B}$ is a function $\tau \colon \mathbb{R}^+_0 \to \mathbb{R}^+_0$ defined as $\tau(r) = \frac{1}{2} \sup_{\|f\|=\|g\|=1} (\|f + rg\| + \|f - rg\|) - 1$, where $f, g \in \mathcal{B}$. The Banach space $\mathcal{B}$ is said to be uniformly smooth if $\tau(r) \in o(r)$ as $r \to 0$.*

We need to observe now that every Hilbert space is a uniformly smooth Banach space [12]. For the sake of completeness, we provide a proof of this proposition in Appendix B.

**Proposition 2.** *For any Hilbert space the modulus of smoothness is equal to $\tau(r) = \sqrt{1 + r^2} - 1$.*

Having shown that Hilbert spaces are uniformly smooth Banach spaces, we proceed with two results giving a convergence rate of an $\epsilon$-greedy incremental sequence of functions. What is interesting about these results is the fact that a feature does not need to match exactly the residual function in a greedy descent step (Section 2.1); it is only required that condition (*ii*) from the next theorem is satisfied.

**Theorem 3.** *[Donahue et al., 12] Let $\mathcal{B}$ be a uniformly smooth Banach space having modulus of smoothness $\tau(u) \leq \gamma u^t$, with $\gamma$ being a constant and $t > 1$. Let $S$ be a bounded subset of $\mathcal{B}$ and let $f \in \overline{\mathrm{co}\,(S)}$. Let $K > 0$ be chosen such that $\|f - g\| \leq K$ for all $g \in S$, and let $\epsilon > 0$ be a fixed slack value. If the sequences $\{f_n\}_{n \geq 1} \subset \mathrm{co}\,(S)$ and $\{g_n\}_{n \geq 1} \subset S$ are chosen recursively so that: (i) $f_1 \in S$, (ii) $F_n(g_n - f) \leq {}^{2\gamma\left((K+\epsilon)^t - K^t\right)} / {n^{t-1}} \|f_n - f\|^{t-1}$, and (iii) $f_{n+1} = {}^n / {n+1}\, f_n + {}^1 / {n+1}\, g_n$, where $F_n$ is the peak functional of $f_n - f$, then it holds $\|f_n - f\| \leq \frac{(2\gamma t)^{1/t}(K+\epsilon)}{n^{1-1/t}} \left[ 1 + \frac{(t-1)\log_2 n}{2tn} \right]^{1/t}.$*

The following corollary gives a convergence rate for an $\epsilon$-greedy incremental sequence of functions constructed according to Theorem 3 with respect to $f_\rho \in \overline{\mathrm{co}\,(\mathcal{F}_\Theta)}$. As this result (a proof is given in Appendix B) holds for all such sequences of functions, it also holds for our constructive procedure.

**Corollary 4.** *Let $\{f_n\}_{n \geq 1} \subset \mathrm{co}\,(\mathcal{F}_\Theta)$ be an $\epsilon$-greedy incremental sequence of functions constructed according to the procedure described in Theorem 3 with respect to a function $f \in \overline{\mathrm{co}\,(\mathcal{F}_\Theta)}$. Then, it holds $\|f_n - f\|_\rho \leq \frac{(K+\epsilon)\sqrt{2 + \log_2 n / 2n}}{\sqrt{n}}.$*

## 2.4 Generalization bound

In step $t + 1$ of the empirical residual fitting, based on a sample $\{(x_i, y_i - f_t(x_i))\}_{i=1}^m$, the approach selects a ridge-wave function from $\mathcal{F}_\Theta$ that approximates well the residual function $(f_\rho - f_t)$. In the last section, we have specified in which cases such ridge-wave functions can be constructed and provided a convergence rate for this constructive procedure. As the convergence result is not limited to target functions from $\mathcal{F}_\Theta$ and $\mathrm{co}\,(\mathcal{F}_\Theta)$, we give a bound on the generalization error for hypotheses from $\mathcal{F} = \overline{\mathrm{co}\,(\mathcal{F}_\Theta)}$, where the closure is taken with respect to $\mathcal{C}(X)$.

Before we give a generalization bound, we show that our hypothesis space $\mathcal{F}$ is a convex and compact set of functions. The choice of a compact hypothesis space is important because it guarantees that a minimizer of the expected squared error $\mathcal{E}_\rho$ and its empirical counterpart $\mathcal{E}_{\mathbf{z}}$ exists. In particular, a continuous function attains its minimum and maximum value on a compact set and this guarantees the existence of minimizers of $\mathcal{E}_\rho$ and $\mathcal{E}_{\mathbf{z}}$. Moreover, for a hypothesis space that is both convex and compact, the minimizer of the expected squared error is *unique as an element of* $\mathcal{L}^2_\rho(X)$. A simple proof of the uniqueness of such a minimizer in $\mathcal{L}^2_\rho(X)$ and the continuity of the functionals $\mathcal{E}_\rho$ and $\mathcal{E}_{\mathbf{z}}$ can be found in [9]. For the sake of completeness, we provide a proof in Appendix A as Proposition A.2. The following proposition (a proof is given in Appendix B) shows that our hypothesis space is a convex and compact subset of the metric space $\mathcal{C}(X)$.

---

**Algorithm 1** GREEDYDESCENT

---

**Input:** sample $\mathbf{z} = \{(x_i, y_i)\}_{i=1}^{s}$, initial estimates at sample points $\{f_{0,i}\}_{i=1}^{s}$, ridge basis function $\phi$, maximum number of descent steps $p$, regularization parameter $\lambda$, and precision $\epsilon$

1:  $W \leftarrow \emptyset$
2:  **for** $k = 1, 2, \ldots, p$ **do**
3:      $w_k, c_k \leftarrow \arg\min_{w, c=(c', c'')} \sum_{i=1}^{s} \left(c' f_{k-1,i} + c'' \phi\left(w^\top x_i\right) - y_i\right)^2 + \lambda \Omega\left(c, w\right)$
4:      $W \leftarrow W \cup \{w_k\}$ and $f_{k,i} \leftarrow c_k' f_{k-1,i} + c_k'' \phi\left(w_k^\top x_i\right), i = 1, \ldots, s$
5:      **if** $|\mathcal{E}_{\mathbf{z}}(f_k) - \mathcal{E}_{\mathbf{z}}(f_{k-1})| / \max\{\mathcal{E}_{\mathbf{z}}(f_k), \mathcal{E}_{\mathbf{z}}(f_{k-1})\} < \epsilon$ **then** EXIT FOR LOOP **end if**
6:  **end for**
7:  **return** $W$

---

**Proposition 5.** *The hypothesis space $\mathcal{F}$ is a convex and compact subset of the metric space $\mathcal{C}(X)$. Moreover, the elements of this hypothesis space are Lipschitz continuous functions.*

Having established that the hypothesis space is a compact set, we can now give a generalization bound for learning with this hypothesis space. The fact that the hypothesis space is compact implies that it is also a totally bounded set [27], i.e., for all $\epsilon > 0$ there exists a finite $\epsilon$-net of $\mathcal{F}$. This, on the other hand, allows us to derive a sample complexity bound by using the $\epsilon$-covering number of a space as a measure of its capacity [21]. The following theorem and its corollary (proofs are provided in Appendix B) give a generalization bound for learning with the hypothesis space $\mathcal{F}$.

**Theorem 6.** *Let $M > 0$ such that, for all $f \in \mathcal{F}$, $|f(x) - y| \leq M$ almost surely. Then, for all $\epsilon > 0$*

$$\mathbb{P}\left[\mathcal{E}_\rho\left(f_{\mathbf{z}}\right) - \mathcal{E}_\rho\left(f^*\right) \leq \epsilon\right] \geq 1 - \mathcal{N}\left(\mathcal{F}, {\epsilon}/{24M}, \|\cdot\|_\infty\right) \exp\left(-{m\epsilon}/{288M^2}\right),$$

*where $f_{\mathbf{z}}$ and $f^*$ are the minimizers of $\mathcal{E}_{\mathbf{z}}$ and $\mathcal{E}_\rho$ on the set $\mathcal{F}$, $\mathbf{z} \in Z^m$, and $\mathcal{N}\left(\mathcal{F}, \epsilon, \|\cdot\|_\infty\right)$ denotes the $\epsilon$-covering number of $\mathcal{F}$ w.r.t. $\mathcal{C}(X)$.*

**Corollary 7.** *For all $\epsilon > 0$ and all $\delta > 0$, with probability $1 - \delta$, a minimizer of the empirical squared error on the hypothesis space $\mathcal{F}$ is $(\epsilon, \delta)$-consistent when the number of samples $m \in \Omega\left(r\left(Rs + t\right) L_\phi \frac{1}{\epsilon^2} + \frac{1}{\epsilon} \ln \frac{1}{\delta}\right)$. Here, $R$ is the radius of a ball containing the set of instances $X$ in its interior, $L_\phi$ is the Lipschitz constant of a function $\phi$, and $r$, $s$, and $t$ are hyperparameters of $\mathcal{F}_\Theta$.*

## 2.5 Algorithm

Algorithm 1 is a pseudo-code description of the outlined approach. To construct a feature space with a good set of linear hypotheses the algorithm takes as input a set of labeled examples and an initial empirical estimate of the target function. A dictionary of features is specified with a ridge basis function and the smoothness of individual features is controlled with a regularization parameter. Other parameters of the algorithm are the maximum allowed number of descent steps and a precision term that defines the convergence of the descent. As outlined in Sections 2.1 and 2.3, the algorithm works by selecting a feature that matches the residual function at the current estimate of the target function. For each selected feature the algorithm also chooses a suitable learning rate and performs a functional descent step (note that we are inferring the learning rate instead of setting it to $1/n+1$ as in Theorem 3). To avoid solving these two problems separately, we have coupled both tasks into a single optimization problem (line 3) – we fit a linear model to a feature representation consisting of the current empirical estimate of the target function and a ridge function parameterized with a $d$-dimensional vector $w$. The regularization term $\Omega$ is chosen to control the smoothness of the new feature and avoid over-fitting. The optimization problem over the coefficients of the linear model and the spectrum of the ridge basis function is solved by casting it as a hyperparameter optimization problem [20]. For the sake of completeness, we have provided a detailed derivation in Appendix C.

While the hyperparameter optimization problem is in general non-convex, Theorem 3 indicates that a globally optimal solution is not (necessarily) required and instead specifies a weaker condition. To account for the non-convex nature of the problem and compensate for the sequential generation of features, we propose to parallelize the feature construction process by running several instances of the greedy descent simultaneously. A pseudo-code description of this parallelized approach is given in Algorithm 2. The algorithm takes as input parameters required for running the greedy descent and some parameters specific to the parallelization scheme – number of data passes and available machines/cores, regularization parameter for the fitting of linear models in the constructed feature space, and cut-off parameter for the elimination of redundant features. The whole process is started by adding a bias feature and setting the initial empirical estimates at sample points to the mean value of the outputs (line 1). Following this, the algorithm mimics stochastic gradient descent and makes

---

**Algorithm 2** GREEDY FEATURE CONSTRUCTION (GFC)

---

**Input:** sample $\mathbf{z} = \{(x_i, y_i)\}_{i=1}^m$, ridge basis function $\phi$, number of data passes $T$, maximum number of greedy descent steps $p$, number of machines/cores $M$, regularization parameters $\lambda$ and $\nu$, precision $\epsilon$, and feature cut-off threshold $\eta$

1:   $W \leftarrow \{\mathbf{0}\}$ and $f_{0,k} \leftarrow \frac{1}{m}\sum_{i=1}^m y_i, k = 1, \ldots, m$
2:   **for** $i = 1, \ldots, T$ **do**
3:      **for** $j = 1, 2, \ldots, M$ **parallel do**
4:        $S_j \sim \mathcal{U}_{\{1,2,\ldots,m\}}$ and $W \leftarrow W \cup \text{GREEDYDESCENT}\left(\{(x_k, y_k)\}_{k \in S_j}, \{f_{i-1,k}\}_{k \in S_j}, \phi, p, \lambda, \epsilon\right)$
5:      **end for**
6:      $a^* \leftarrow \arg\min_a \sum_{k=1}^m \left(\sum_{l=1}^{|W|} a_l \phi\left(w_l^\top x_k\right) - y_k\right)^2 + \nu \|a\|_2^2$
7:      $W \leftarrow W \setminus \{w_l \in W \mid |a_l^*| < \eta, 1 \le l \le |W|\}$ and $f_{i,k} \leftarrow \sum_{l=1}^{|W|} a_l^* \phi\left(w_l^\top x_k\right), k = 1, \ldots, m$
8:   **end for**
9:   **return** $(W, a^*)$

---

a specified number of passes through the data (line 2). In the first step of each pass, the algorithm performs greedy functional descent in parallel using a pre-specified number of machines/cores $M$ (lines 3-5). This step is similar to the splitting step in parallelized stochastic gradient descent [32]. Greedy descent is performed on each of the machines for a maximum number of iterations $p$ and the estimated parameter vectors are added to the set of constructed features $W$ (line 4). After the features have been learned the algorithm fits a linear model to obtain the amplitudes (line 6). To fit a linear model, we use least square regression penalized with the $l_2$-norm because it can be solved in a closed form and cross-validation of the capacity parameter involves optimizing a 1-dimensional objective function [20]. Fitting of the linear model can be understood as averaging of the greedy approximations constructed on different chunks of the data. At the end of each pass, the empirical estimates at sample points are updated and redundant features are removed (line 7).

One important detail in the implementation of Algorithm 1 is the data splitting between the training and validation samples for the hyperparameter optimization. In particular, during the descent we are more interested in obtaining a good spectrum than the amplitude because a linear model will be fit in Algorithm 2 over the constructed features and the amplitude values will be updated. For this reason, during the hyperparameter optimization over a $k$-fold splitting in Algorithm 1, we choose a single fold as the training sample and a batch of folds as the validation sample.

## 3   Experiments

In this section, we assess the performance of our approach (see Algorithm 2) by comparing it to other feature construction approaches on synthetic and real-world data sets. We evaluate the effectiveness of the approach with the set of cosine-wave features introduced in Section 2.2. For this set of features, our approach is directly comparable to random Fourier features [26] and á la carte [31]. The implementation details of the three approaches are provided in Appendix C. We address here the choice of the regularization term in Algorithm 1: To control the smoothness of newly constructed features, we penalize the objective in line 3 so that the solutions with the small $\mathcal{L}_\rho^2\left(X\right)$ norm are preferred. For this choice of regularization term and cosine-wave features, we empirically observe that the optimization objective is almost exclusively penalized by the $l_2$ norm of the coefficient vector $c$. Following this observation, we have simulated the greedy descent with $\Omega\left(c, w\right) = \|c\|_2^2$.

We now briefly describe the data sets and the experimental setting. The experiments were conducted on three groups of data sets. The first group contains four UCI data sets on which we performed parameter tuning of all three algorithms (Table 1, data sets 1-4). The second group contains the data sets with more than 5000 instances available from Luís Torgo [28]. The idea is to use this group of data sets to test the generalization properties of the considered algorithms (Table 1, data sets 5-10). The third group contains two artificial and very noisy data sets that are frequently used in regression tree benchmark tests. For each considered data set, we split the data into 10 folds; we refer to these splits as the outer cross-validation folds. In each step of the outer cross-validation, we use nine folds as the training sample and one fold as the test sample. For the purpose of the hyperparameter tuning we split the training sample into five folds; we refer to these splits as the inner cross-validation folds. We run all algorithms on identical outer cross-validation folds and construct feature representations with 100 and 500 features. The performance of the algorithms is assessed by comparing the root mean squared error of linear ridge regression models trained in the constructed feature spaces and the average time needed for the outer cross-validation of one fold.

Table 1: To facilitate the comparison between data sets we have normalized the outputs so that their range is one. The accuracy of the algorithms is measured using the root mean squared error, multiplied by 100 to mimic percentage error (w.r.t. the range of the outputs). The mean and standard deviation of the error are computed after performing 10-fold cross-validation. The reported walltime is the average time it takes a method to cross-validate one fold. To assess whether a method performs statistically significantly better than the other on a particular data set we perform the paired Welch t-test [29] with $p = 0.05$. The significantly better results for the considered settings are marked in bold.

| DATASET | $m$ | $d$ | $n = 100$ | | | | $n = 500$ | | | |
|---|---|---|---|---|---|---|---|---|---|---|
| | | | GFC | | ALC | | GFC | | ALC | |
| | | | ERROR | WALLTIME | ERROR | WALLTIME | ERROR | WALLTIME | ERROR | WALLTIME |
| parkinsons tm (total) | 5875 | 21 | **2.73** (±0.19) | 00 : 03 : 49 | **0.78** (±0.13) | 00 : 05 : 19 | **2.20** (±0.27) | 00 : 04 : 15 | **0.31** (±0.17) | 00 : 27 : 15 |
| ujindoorloc (latitude) | 21048 | 527 | **3.17** (±0.15) | 00 : 21 : 39 | 6.19 (±0.76) | 01 : 21 : 58 | **3.04** (±0.19) | 00 : 36 : 49 | 6.99 (±0.97) | 02 : 23 : 15 |
| ct-slice | 53500 | 380 | **2.93** (±0.10) | 00 : 52 : 05 | 3.82 (±0.64) | 03 : 31 : 25 | **2.59** (±0.10) | 01 : 24 : 41 | 2.73 (±0.29) | 06 : 11 : 12 |
| Year Prediction MSD | 515345 | 90 | 10.06 (±0.09) | 01 : 20 : 12 | **9.94** (±0.08) | 05 : 29 : 14 | 10.01 (±0.08) | 01 : 30 : 28 | **9.92** (±0.07) | 11 : 58 : 41 |
| delta-ailerons | 7129 | 5 | 3.82 (±0.24) | 00 : 01 : 23 | 3.73 (±0.20) | 00 : 05 : 13 | 3.79 (±0.25) | 00 : 01 : 57 | 3.73 (±0.24) | 00 : 25 : 14 |
| kinematics | 8192 | 8 | 5.18 (±0.09) | 00 : 04 : 02 | 5.03 (±0.23) | 00 : 11 : 28 | 4.65 (±0.11) | 00 : 04 : 44 | 5.01 (±0.76) | 00 : 38 : 53 |
| cpu-activity | 8192 | 21 | 2.65 (±0.12) | 00 : 04 : 23 | 2.68 (±0.27) | 00 : 09 : 24 | 2.60 (±0.16) | 00 : 04 : 24 | 2.62 (±0.15) | 00 : 25 : 13 |
| bank | 8192 | 32 | 9.83 (±0.27) | 00 : 01 : 39 | 9.84 (±0.30) | 00 : 12 : 48 | 9.83 (±0.30) | 00 : 02 : 01 | 9.87 (±0.42) | 00 : 49 : 48 |
| pumadyn | 8192 | 32 | 3.44 (±0.10) | 00 : 02 : 24 | **3.24** (±0.07) | 00 : 13 : 17 | **3.30** (±0.06) | 00 : 02 : 27 | 3.42 (±0.15) | 00 : 57 : 33 |
| delta-elevators | 9517 | 6 | 5.26 (±0.17) | 00 : 00 : 57 | 5.28 (±0.18) | 00 : 07 : 07 | 5.24 (±0.17) | 00 : 01 : 04 | 5.23 (±0.18) | 00 : 32 : 30 |
| ailerons | 13750 | 40 | 4.67 (±0.18) | 00 : 02 : 56 | 4.89 (±0.43) | 00 : 16 : 34 | 4.51 (±0.12) | 00 : 02 : 11 | 4.77 (±0.40) | 01 : 05 : 07 |
| pole-telecom | 15000 | 26 | 7.34 (±0.29) | 00 : 10 : 45 | 7.16 (±0.55) | 00 : 20 : 34 | 5.55 (±0.15) | 00 : 11 : 37 | 5.20 (±0.51) | 01 : 39 : 22 |
| elevators | 16599 | 18 | 3.34 (±0.08) | 00 : 03 : 16 | 3.37 (±0.55) | 00 : 21 : 20 | 3.12 (±0.20) | 00 : 04 : 06 | 3.13 (±0.24) | 01 : 20 : 58 |
| cal-housing | 20640 | 8 | **11.55** (±0.24) | 00 : 05 : 49 | 12.69 (±0.47) | 00 : 11 : 14 | **11.17** (±0.25) | 00 : 06 : 16 | 12.70 (±1.01) | 01 : 01 : 37 |
| breiman | 40768 | 10 | **4.01** (±0.03) | 00 : 02 : 46 | 4.06 (±0.04) | 00 : 13 : 52 | **4.01** (±0.03) | 00 : 03 : 04 | 4.03 (±0.03) | 01 : 04 : 16 |
| friedman | 40768 | 10 | 3.29 (±0.09) | 00 : 06 : 07 | 3.37 (±0.46) | 00 : 18 : 43 | **3.16** (±0.03) | 00 : 07 : 04 | 3.25 (±0.09) | 01 : 39 : 37 |

An extensive summary containing the results of experiments with the random Fourier features approach (corresponding to Gaussian, Laplace, and Cauchy kernels) and different configurations of á la carte is provided in Appendix D. As the best performing configuration of á la carte on the development data sets is the one with $Q = 5$ components, we report in Table 1 the error and walltime for this configuration. From the walltime numbers we see that our approach is in both considered settings – with 100 and 500 features – always faster than á la carte. Moreover, the proposed approach is able to generate a feature representation with 500 features in less time than required by á la carte for a representation of 100 features. In order to compare the performance of the two methods with respect to accuracy, we use the Wilcoxon signed rank test [30, 11]. As our approach with 500 features is on all data sets faster than á la carte with 100 features, we first compare the errors obtained in these experiments. For 95% confidence, the threshold value of the Wilcoxon signed rank test with 16 data sets is $T = 30$ and from our results we get the T-value of 28. As the T-value is below the threshold, our algorithm can with 95% confidence generate in less time a statistically significantly better feature representation than á la carte. For the errors obtained in the settings where both methods have the same number of features, we obtain the T-values of 60 and 42. While in the first case for the setting with 100 features the test is inconclusive, in the second case our approach is with 80% confidence statistically significantly more accurate than á la carte. To evaluate the performance of the approaches on individual data sets, we perform the paired Welch t-test [29] with $p = 0.05$. Again, the results indicate a good/competitive performance of our algorithm compared to á la carte.

## 4   Discussion

In this section, we discuss the advantages of the proposed method over the state-of-the-art baselines in learning fast shift-invariant kernels and other related approaches.

**Flexibility.** The presented approach is a highly flexible supervised feature construction method. In contrast to an approach based on random Fourier features [26], the proposed method does not require a spectral measure to be specified a priori. In the experiments (details can be found in Appendix D), we have demonstrated that the choice of spectral measure is important as, for the considered measures (corresponding to Gaussian, Laplace, and Cauchy kernels), the random Fourier features approach is outperformed on all data sets. The second competing method, á la carte, is more flexible when it comes to the choice of spectral measure and works by approximating it with a mixture of Gaussians. However, the number of components and features per component needs to be specified beforehand or cross-validated. In contrast, our approach mimics functional gradient descent and can be simulated without specifying the size of the feature representation beforehand. Instead, a stopping criteria (see, e.g., Algorithm 1) based on the successive decay of the error can be devised. As a result, the proposed approach terminates sooner than the alternative approaches for simple concepts/hypothesis. The method is also easy to implement (for the sake of completeness, the hyperparameter gradients are provided in Appendix C.1) and allows us to extend the existing feature representation without complete re-training of the model. We note that the approaches based on random Fourier

features are also simple to implement and can be re-trained efficiently with the increase in the number of features [10]. Á la carte, on the other hand, is less flexible in this regard – due to the number of hyperparameters and the complexity of gradients it is not straightforward to implement this method.

**Scalability.** The fact that our greedy descent can construct a feature in time linear in the number of instances $m$ and dimension of the problem $d$ makes the proposed approach highly scalable. In particular, the complexity of the proposed parallelization scheme is dominated by the cost of fitting a linear model and the whole algorithm runs in time $\mathcal{O}\left(T\left(n^3 + n^2 m + nmd\right)\right)$, where $T$ denotes the number of data passes (i.e., linear model fits) and $n$ number of constructed features. To scale this scheme to problems with millions of instances, it is possible to fit linear models using the parallelized stochastic gradient descent [32]. As for the choice of $T$, the standard setting in simulations of stochastic gradient descent is 5-10 data passes. Thus, the presented approach is quite robust and can be applied to large scale data sets. In contrast to this, the cost of performing a gradient step in the hyperparameter optimization of á la carte is $\mathcal{O}\left(n^3 + n^2 m + nmd\right)$. In our empirical evaluation using an implementation with 10 random restarts, the approach needed at least 20 steps per restart to learn an accurate model. The required number of gradient steps and the cost of computing them hinder the application of á la carte to large scale data sets. In learning with random Fourier features which also run in time $\mathcal{O}\left(n^3 + n^2 m + nmd\right)$, the main cost is the fitting of linear models – one for each pair of considered spectral and regularization parameters.

**Other approaches.** Beside fast kernel learning approaches, the presented method is also related to neural networks parameterized with a single hidden layer. These approaches can be seen as feature construction methods jointly optimizing over the whole feature representation. A detailed study of the approximation properties of a hypothesis space of a single layer network with the sigmoid ridge function has been provided by Barron [4]. In contrast to these approaches, we construct features incrementally by fitting residuals and we do this with a set of non-monotone ridge functions as a dictionary of features. Regarding our generalization bound, we note that the past work on single layer neural networks contains similar results but in the context of monotone ridge functions [1].

As the goal of our approach is to construct a feature space for which linear hypotheses will be of sufficient capacity, the presented method is also related to linear models working with low-rank kernel representations. For instance, Fine and Scheinberg [14] investigate a training algorithm for SVMs using low-rank kernel representations. The difference between our approach and this method is in the fact that the low-rank decomposition is performed without considering the labels. Side knowledge and labels are considered by Kulis et al. [22] and Bach and Jordan [3] in their approaches to construct a low-rank kernel matrix. However, these approaches are not selecting features from a set of ridge functions, but find a subspace of a preselected kernel feature space with a good set of hypothesis.

From the perspective of the optimization problem considered in the greedy descent (Algorithm 1) our approach can be related to single index models (SIM) where the goal is to learn a regression function that can be represented as a single monotone ridge function [19, 18]. In contrast to these models, our approach learns target/regression functions from the closure of the convex hull of ridge functions. Typically, these target functions cannot be written as single ridge functions. Moreover, our ridge functions do not need to be monotone and are more general than the ones considered in SIM models.

In addition to these approaches and considered baseline methods, the presented feature construction approach is also related to methods optimizing expected loss functions using functional gradient descent [23]. However, while Mason et al. [23] focus on classification problems and hypothesis spaces with finite VC dimension, we focus on the estimation of regression functions in spaces with infinite VC dimension (e.g., see Section 2.2). In contrast to that work, we provide a convergence rate for our approach. Similarly, Friedman [15] has proposed a gradient boosting machine for greedy function estimation. In their approach, the empirical functional gradient is approximated by a weak learner which is then combined with previously constructed learners following a *stagewise* strategy. This is different from the *stepwise* strategy that is followed in our approach where previously constructed estimators are readjusted when new features are added. The approach in [15] is investigated mainly in the context of regression trees, but it can be adopted to feature construction. To the best of our knowledge, theoretical and empirical properties of this approach in the context of feature construction and shift-invariant reproducing kernel Hilbert spaces have not been considered so far.

**Acknowledgment:** We are grateful for access to the University of Nottingham High Performance Computing Facility. A part of this work was also supported by the German Science Foundation (grant number GA 1615/1-1).

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
