[Supplementary Material]

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

# A Preliminaries

**Definition A.1.** *A symmetric function $k\colon X \times X \to \mathbb{R}$ is a positive definite kernel on $X$ if, for all $n \in \mathbb{N}$, $x_1, \ldots, x_n \in X$, and $c_1, \ldots, c_n \in \mathbb{R}$, it follows that $\sum_{i,j=1}^{n} c_i c_j k\,(x_i, x_j) \geq 0$.*

**Definition A.2.** *Let $D \subset \mathbb{R}^d$ be an open set. A positive definite kernel $k\colon D \times D \to \mathbb{R}$ is called shift-invariant if there exists a function $s\colon D \to \mathbb{R}$ such that $k\,(x, y) = s\,(x - y)$, for all $x, y \in D$. The function $s$ is said to be of positive type.*

**Definition A.3.** *A reproducing kernel Hilbert space $\mathcal{H}$ on a non-empty set $X$ is the Hilbert space of functions $f\colon X \to \mathbb{R}$ such that there exists a unique element $\mathrm{e}_x \in \mathcal{H}$ satisfying the reproducing property $f\,(x) = \langle f, \mathrm{e}_x\rangle_{\mathcal{H}}$ for all $f \in \mathcal{H}$. For a reproducing kernel Hilbert space $\mathcal{H}$ the function $k\,(x, y) = \mathrm{e}_x\,(y)$ is a positive definite kernel. A unique reproducing kernel Hilbert space $\mathcal{H}_k$ corresponds to every positive definite kernel $k$ [2].*

**Theorem A.1.** *[Bochner, 7] The Fourier transform of a bounded positive measure on $\mathbb{R}^d$ is a continuous function of positive type. Conversely, any function of positive type is the Fourier transform of a bounded positive measure.*

In other words, for a shift-invariant kernel $k$ it holds

$$k\,(x, y) = s\,(x - y) = \int_{\mathbb{R}^d} \exp\,(-i\,\langle w, x - y\rangle)\ \mathrm{d}\mu\,(w),$$

where $\mu$ is a positive and bounded measure. As $k\,(x, y)$ is a real function in both arguments, the complex part in the integral on the right hand-side is equal to zero, and we have

$$k\,(x, y) = 2 \int \cos\,(w^\top x + b) \cos\,(w^\top y + b)\ \mathrm{d}\hat{\mu}\,(w, b),$$

where $b \sim \mathcal{U}_{[-\pi, \pi]}$ and $\hat{\mu}\,(w, b) = {}^{\mu(w)}\!/{}_{2\pi}$. Hence, the kernel value at $(x, y)$ can be approximated by the Monte-Carlo estimate of the dot product [26].

**Proposition A.2.** *[Cucker and Smale, 9] Let $\mathcal{K}$ be a convex and compact subset of $\mathcal{C}\,(X)$. Then there exists a function in $\mathcal{C}\,(X)$ with a minimal distance to $f_\rho$ in $\mathcal{L}_\rho^2\,(X)$. Moreover, this function is unique as an element of $\mathcal{L}_\rho^2\,(X)$.*

*Proof.* From the compactness of the subspace it follows that a minimizer exists. However, it does not have to be unique. Let $f_1$ and $f_2$ be two minimizers and let $s = \{\alpha f_1 + (1 - \alpha)\, f_2 \mid 0 \leq \alpha \leq 1\}$ be the line segment connecting these two points. As the subspace $\mathcal{K}$ is convex, then the segment $s$ is contained within $\mathcal{K}$. Furthermore, for all $f \in s$, it holds $\|f_1 - f_\rho\|_\rho = \|f_2 - f_\rho\|_\rho \leq \|f - f_\rho\|_\rho$. From the first inequality we have

$$\langle f_\rho - f_1, f - f_1\rangle_\rho + \langle f_\rho - f_1, f_\rho - f\rangle_\rho \leq \|f_\rho - f\|_\rho^2 \Rightarrow \langle f_\rho - f_1, f - f_1\rangle_\rho \leq \langle f_1 - f, f_\rho - f\rangle_\rho.$$

Similarly, from the second inequality we get

$$\langle f_\rho - f_2, f - f_2\rangle_\rho \leq \langle f_2 - f, f_\rho - f\rangle_\rho.$$

As the cosine is decreasing function over $[0, \pi]$, it follows that $\angle f_\rho f_1 f \geq \angle f_\rho f f_1$ and $\angle f_\rho f_2 f \geq \angle f_\rho f f_2$ for all $f \in s$. Hence, if $f_1 \neq f_2$ then the angles $\angle f_\rho f_1 f$ and $\angle f_\rho f_2 f$ are obtuse. As there does not exist a triangle with two obtuse angles, this is impossible and $f_1 = f_2$. $\qquad\square$

**Proposition A.3.** *[Cucker and Smale, 9] Let $f_1, f_2 \in \mathcal{C}\,(X)$, $M \in \mathbb{R}_+$, and $|f_i\,(x) - y| \leq M$ on a set $U \subset Z$ of full measure for $i = 1, 2$. Then for all $\mathbf{z} \in U^m$ functions $\mathcal{E}_\rho$ and $\mathcal{E}_\mathbf{z}$ are Lipschitz continuous on the metric space $\mathcal{C}\,(X)$.*

*Proof.* We have that

$$\left|(f_1\,(x) - y)^2 - (f_2\,(x) - y)^2\right| = |f_1\,(x) - f_2\,(x)|\,|f_1\,(x) - y + f_2\,(x) - y| \leq 2M\,\|f_1 - f_2\|_\infty$$

and the claim follows from this inequality. $\qquad\square$

**Definition A.4.** *The space is called centralizable if in it, for any open set $U$ of diameter $2d$, there exists a point $x_0$ from which any point $x$ is at a distance no greater than $d$.*

**Theorem A.4.** *[Kolmogorov and Tikhomirov, 21] Let $S$ be a connected totally bounded set which is contained in a centralizable space and let $\mathrm{Lip}_1(S)$ be a set of bounded $1$-Lipschitz continuous functions on $S$. If all functions from $\mathrm{Lip}_1(S)$ are bounded by a constant $C > 0$, then it holds*

$$\mathcal{N}\left(\mathrm{Lip}_1(S), \epsilon, \|\cdot\|_\infty\right) \leq 2^{\mathcal{N}\left(S, \frac{\epsilon}{2}, \|\cdot\|_2\right)} \left(2\left\lceil \frac{2C}{\epsilon} \right\rceil + 1\right).$$

*Proof.* As the set $S$ is totally bounded, then for all $\epsilon > 0$ there exists a finite $\epsilon$-cover of $S$. Let $\{U\}_{i=1}^n$ denote the $\frac{\epsilon}{2}$-cover of the set $S$ and let $x_i$ be the center of the set $U_i$. Let $f \in \mathrm{Lip}_1(S)$ and $\hat{f}$ be an approximation of $f$. Define $\hat{f}$ over the set $U_1$ as the number $\left\lceil \frac{2f(x_1)}{\epsilon} \right\rceil \frac{\epsilon}{2}$. Then, for all $x \in U_1$

$$\left| f(x) - \hat{f}(x) \right| = \left| f(x) - \hat{f}(x_1) \right| \leq \left| f(x) - f(x_1) + \frac{\epsilon}{2} \right| \leq \epsilon.$$

Setting $x = x_1$ we see that $\left| f(x_1) - \hat{f}(x_1) \right| \leq \frac{\epsilon}{2}$.

On the other hand, for the center of the set $U_i$ that is adjacent to $U_1$, $U_i \cap U_1 \neq \emptyset$, it holds

$$\left| f(x_i) - \hat{f}(x_1) \right| \leq |f(x_i) - f(x_1)| + \left| f(x_1) - \hat{f}(x_1) \right| \leq \frac{\epsilon}{2} + \frac{\epsilon}{2} = \epsilon.$$

This means that knowing the value at the center of $U_1$ with precision $\frac{\epsilon}{2}$ suffices to approximate with precision $\epsilon$ the value at the centers of neighbouring sets in the cover. From here it follows that by taking $\hat{f}(x) = \hat{f}(x_1) \pm \frac{\epsilon}{2}$ for all $x \in U_i$, such that $U_1$ and $U_i$ are adjacent, we can approximate $\left| f(x_i) - \hat{f}(x_i) \right|$ with precision $\frac{\epsilon}{2}$. As the space $S$ is connected it is possible to connect any two non-adjacent sets $U_i$ and $U_j$ by a sequence of intersecting sets $U_k$. Hence, we can construct the entire functional $\hat{f}$ in this way and approximate the function $f$ such that $\left\| f - \hat{f} \right\|_\infty \leq \epsilon$.

Now, covering the range of these functions, $[-C, C]$, with $\epsilon$-intervals we see that it is sufficient to take $2\left\lceil \frac{2C}{\epsilon} \right\rceil + 1$ numbers as the center-values at $x_1$. For each of the sets in the $\frac{\epsilon}{2}$-cover we have two choices and thus the $\epsilon$-covering number of $\mathrm{Lip}_1(S)$ is not greater than $2^{\mathcal{N}\left(S, \frac{\epsilon}{2}, \|\cdot\|\right)} \left(2\left\lceil \frac{2C}{\epsilon} \right\rceil + 1\right)$. $\square$

**Theorem A.5.** *[Cucker and Smale, 9] Let $\mathcal{K}$ be a compact and convex subset of $\mathcal{C}(X)$ and let $M > 0$ be a finite constant such that for all $f \in \mathcal{K}$, $|f(x) - y| \leq M$ almost everywhere. Then, for all $\epsilon > 0$,*

$$\mathbb{P}_{\mathbf{z} \in Z^m}\left[\mathcal{E}_\rho(f_{\mathbf{z}}) - \mathcal{E}_\rho(f_{\mathcal{K}}) \leq \epsilon\right] \geq 1 - \mathcal{N}\left(\mathcal{K}, \frac{\epsilon}{24M}, \|\cdot\|_\infty\right) \exp\left(-\frac{m\epsilon}{288M^2}\right),$$

*where $f_{\mathbf{z}}$ and $f_{\mathcal{K}}$ are the minimizers of $\mathcal{E}_{\mathbf{z}}$ and $\mathcal{E}_\rho$ over $\mathcal{K}$.*

On the other hand, for the approximation over a compact space only, the following theorem holds.

**Theorem A.6.** *[Cucker and Smale, 9] Let $\mathcal{K}$ be a compact subset of $\mathcal{C}(X)$ and let $M > 0$ be a finite constant such that, for all $f \in \mathcal{K}$, $|f(x) - y| \leq M$ almost everywhere. Then, for all $\epsilon > 0$,*

$$\mathbb{P}_{\mathbf{z} \in Z^m}\left[\sup_{f \in \mathcal{K}} |\mathcal{E}_\rho(f) - \mathcal{E}_{\mathbf{z}}(f)| \leq \epsilon\right] \geq 1 - 2\mathcal{N}\left(\mathcal{K}, \frac{\epsilon}{8M}, \|\cdot\|_\infty\right) \exp\left(-\frac{m\epsilon^2}{4\left(2\sigma^2 + \frac{1}{3}M^2\epsilon\right)}\right),$$

*where $\sigma^2 = \sup_{f \in \mathcal{K}} \mathrm{Var}_\rho\left[(f(x) - y)^2\right]$.*

**Proposition A.7.** *[Carl and Stephani, 8] Let $\mathbb{E}$ be a finite dimensional Banach space and let $B_R$ be the ball of radius $R$ centered at the origin. Then, for $d = \dim(\mathbb{E})$*

$$\mathcal{N}(B_R, \epsilon, \|\cdot\|) \leq \left(\frac{4R}{\epsilon}\right)^d.$$

# B  Proofs

**Theorem 1.** *Let $\mathcal{H}_k$ be a reproducing kernel Hilbert space corresponding to a continuous shift-invariant and positive definite kernel $k$ defined on a compact set $X$. Let $\mu$ be the positive and bounded spectral measure whose Fourier transform is the kernel $k$. For any probability measure $\rho$ defined on $X$, it is possible to approximate any bounded function $f \in \mathcal{H}_k$ using a convex combination of $n$ ridge-wave functions from $\mathcal{F}_{\cos}$ such that the approximation error in $\|\cdot\|_\rho$ decays with rate $\mathcal{O}\left(1/\sqrt{n}\right)$.*

*Proof.* Let $f \in \mathcal{H}_k$ be any bounded function. From the definition of $\mathcal{H}_k$ it follows that the set $\mathcal{H}_0 = \mathrm{span}\left\{k\left(x,\cdot\right) \mid x \in X\right\}$ is a dense subset of $\mathcal{H}_k$. In other words, for every $\epsilon > 0$ there is a bounded function $g \in \mathcal{H}_0$ such that $\|f - g\|_{\mathcal{H}_k} < \epsilon$.

As feature functions $k\left(x,\cdot\right)$ are continuous and defined on the compact set $X$, they are also bounded. Thus, we can assume that there exists a constant $B > 0$ such that $\sup_{x,y\in X} |k\left(x,y\right)| < B$. From here it follows

$$\|f - g\|_\infty = \sup_{x\in X} \left| \langle f - g, k\left(x,\cdot\right) \rangle_{\mathcal{H}_k} \right| \leq \sqrt{B}\, \|f - g\|_{\mathcal{H}_k}.$$

This means that convergence in $\|\cdot\|_{\mathcal{H}_k}$ implies the uniform convergence. The uniform convergence, on the other hand, implies the convergence in $L^2_\rho$ norm, i.e., for any probability measure $\rho$ on the set $X$, for any $\epsilon > 0$, and for any $f \in \mathcal{H}_k$ there exists $g \in \mathcal{H}_0$ such that

$$\|f - g\|_\rho < \epsilon. \tag{1}$$

The function $g$ is by definition a finite linear combination of feature functions $k\left(x_i,\cdot\right)$ [see, e.g., Chapter 1 in 6] and by Theorem A.1 it can be written as

$$
\begin{aligned}
g\left(x\right) &= \sum_{i=1}^{l} \alpha_i k\left(x_i, x\right) = 2 \int \left( \sum_{i=1}^{l} \alpha_i \cos\left(w^\top x_i + b\right) \right) \cos\left(w^\top x + b\right)\, \mathrm{d}\hat{\mu}\left(w, b\right) \\
&= 2\mu\left(0\right) \int u\left(w, b\right) \cos\left(w^\top x + b\right)\, \mathrm{d}\tilde{\mu}\left(w, b\right),
\end{aligned}
$$

where $\tilde{\mu}$ is a probability measure on $\mathbb{R}^d \times [-\pi, \pi]$, $u\left(w, b\right) = \sum_{i=1}^{l} \alpha_i \cos\left(w^\top x_i + b\right)$, and $\int \mathrm{d}\hat{\mu}\left(w, b\right) = \mu\left(0\right) < \infty$. From the boundedness of $g$, it follows that the function $u$ is bounded for all $w$ and $b$ from the support of $\tilde{\mu}$, i.e., $|u\left(w, b\right)| \leq \sum_{i=1}^{l} |\alpha_i| < \infty$. Denoting with $\gamma\left(w, b\right) = 2\mu\left(0\right) u\left(w, b\right)$, we see that it is sufficient to prove that

$$\mathbb{E}_{\tilde{\mu}(w,b)} \left[ \gamma\left(w, b\right) \cos\left(w^\top x + b\right) \right] \in \overline{\mathrm{co}\left(\mathcal{F}_{\cos}\right)},$$

where the closure is taken with respect to the norm in $L^2_\rho\left(X\right)$. In particular, for a sample $(\mathbf{w}, \mathbf{b}) = \left\{(w_i, b_i)\right\}_{i=1}^{s}$ drawn independently from $\tilde{\mu}$ we have

$$
\mathbb{E}_{(\mathbf{w},\mathbf{b})} \left[ \int \left( g\left(x\right) - \frac{1}{s} \sum_{i=1}^{s} \gamma\left(w_i, b_i\right) \cos\left(w_i^\top x + b_i\right) \right)^2 \mathrm{d}\rho \right] =
$$

$$
\frac{1}{s^2} \int \mathbb{E}_{(\mathbf{w},\mathbf{b})} \left[ \left( \sum_{i=1}^{s} \underbrace{g\left(x\right) - \gamma\left(w_i, b_i\right) \cos\left(w_i^\top x + b_i\right)}_{\xi(x;\, w_i, b_i)} \right)^2 \right] \mathrm{d}\rho =
$$

$$
\frac{1}{s^2} \int \mathbb{E}_{(\mathbf{w},\mathbf{b})} \left[ \left( \sum_{i=1}^{s} \xi\left(x;\, w_i, b_i\right) \right)^2 \right] \mathrm{d}\rho = \frac{1}{s} \int \mathbb{E}_{\tilde{\mu}} \left[ \xi\left(x;\, w, b\right)^2 \right] \mathrm{d}\rho =
$$

$$
\frac{1}{s} \int \mathrm{Var}_{\tilde{\mu}} \left[ g\left(x\right) - \gamma\left(w, b\right) \cos\left(w^\top x + b\right) \right] \mathrm{d}\rho = \frac{1}{s} \int \mathrm{Var}_{\tilde{\mu}} \left[ \gamma\left(w, b\right) \cos\left(w^\top x + b\right) \right] \mathrm{d}\rho.
$$

Note that the third equation follows from the fact that $\xi\left(x;\, w_i, b_i\right)$ are independent and identically distributed random variables and $\mathbb{E}\left[ \xi\left(x;\, w_i, b_i\right) \xi\left(x;\, w_j, b_j\right) \right] = 0$. As established earlier, coefficients $\gamma\left(w, b\right)$ are bounded and, therefore, random variable $\eta_x\left(w, b\right) = \gamma\left(w, b\right) \cos\left(w^\top x + b\right)$ is

bounded, as well. Hence, from $\sup_{w,b} |\eta_x(w,b)| = D < \infty$ it follows that $\mathrm{Var}_{\tilde{\mu}}(\eta_x(w,b)) \le D^2$ and consequently for $g_s(x; (\mathbf{w}, \mathbf{b})) = \frac{1}{s} \sum_{i=1}^{s} \gamma(w_i, b_i) \cos(w_i^\top x + b_i)$ we get

$$\mathbb{E}_{g_s}\left[\|g - g_s\|_\rho^2\right] \le \frac{D^2}{s}. \tag{2}$$

As the expected value of the norm $\|g - g_s\|_\rho$ is bounded by a constant, it follows that there exists a function $g_s$ which can be represented as a convex combination of $s$ ridge-wave functions from $\mathcal{F}_{\cos}$ and for which it holds $\|g - g_s\|_\rho \in \mathcal{O}\left(\frac{1}{\sqrt{s}}\right)$. Moreover, there exists a sequence of functions $\{g_n\}_{n \ge 1}$ converging to $g$ in $\|\cdot\|_\rho$ such that each $g_n$ is a convex combination of $n$ elements from $\mathcal{F}_{\cos}$ and $\|g - g_n\|_\rho \in \mathcal{O}\left(\frac{1}{\sqrt{n}}\right)$.

Hence, we have proved that $g \in \overline{\mathrm{co}\,(\mathcal{F}_{\cos})}$, where the closure is taken with respect to $\|\cdot\|_\rho$. It is then possible to approximate any bounded function $f \in \mathcal{H}_k$ using a convex combination of $n$ ridge-wave functions from $\mathcal{F}_{\cos}$ with the rate $\mathcal{O}\left(\frac{1}{\sqrt{n}}\right)$, i.e., for all $n \in \mathbb{N}$

$$\|f - g_n\|_\rho \le \|f - g\|_\rho + \|g - g_n\|_\rho \in \mathcal{O}\left(\frac{1}{\sqrt{n}}\right).$$

$\square$

**Proposition 2.** *For any Hilbert space the modulus of smoothness is equal to $\tau(r) = \sqrt{1 + r^2} - 1$.*

*Proof.* Expanding norms using the dot product we get

$$2\,(\tau(r) + 1) = \sup_{\|f\| = \|g\| = 1} \left(\sqrt{1 + r^2 + 2r\,\langle f, g\rangle} + \sqrt{1 + r^2 - 2r\,\langle f, g\rangle}\right).$$

Denoting with $u = 1 + r^2$ and $v = 2r\,\langle f, g\rangle$ and using the inequality between arithmetic and quadratic mean we get

$$\sqrt{u + v} + \sqrt{u - v} \le 2\sqrt{\frac{u + v + u - v}{2}} = 2\sqrt{u}.$$

As the equality is attained for $v = 0$ it follows that the modulus of smoothness of a Hilbert space is given by

$$\tau(r) = \sqrt{1 + r^2} - 1.$$

$\square$

**Corollary 4.** *Let $\{f_n\}_{n \ge 1} \subset \mathrm{co}\,(\mathcal{F}_\Theta)$ be an $\epsilon$-greedy incremental sequence of functions constructed according to the procedure described in Theorem 3 with respect to a function $f \in \overline{\mathrm{co}\,(\mathcal{F}_\Theta)}$. Then, it holds $\|f_n - f\|_\rho \le \frac{(K + \epsilon)\sqrt{2 + \log_2 n/2n}}{\sqrt{n}}$.*

*Proof.* As $\mathcal{L}_\rho^2(X)$ is a Hilbert space, it follows from Proposition 2 that the modulus of smoothness of this space is $\tau(r) = \sqrt{1 + r^2} - 1$. While it is straightforward to show that $\sqrt{1 + r^2} \le 1 + r$ for $r \in \mathbb{R}_0^+$, this bound is not tight enough as $r \to 0$. A tighter upper bound for this modulus of smoothness can be derived from the inequality $\sqrt{1 + r^2} \le 1 + \frac{r^2}{2}$. To see that this is a better bound for the case when $r \to 0$, it is sufficient to check that $1 + \frac{r^2}{2} \le 1 + r$ for all $0 \le r \le 2$.

Hence, all conditions of Theorem 3 are satisfied and the claim follows by taking $t = 2$ and $\gamma = \frac{1}{2}$. $\square$

**Proposition 5.** *The hypothesis space $\mathcal{F}$ is a convex and compact subset of the metric space $\mathcal{C}(X)$. Moreover, the elements of this hypothesis space are Lipschitz continuous functions.*

*Proof.* Let $f, g \in \mathcal{F}$. As the hypothesis space $\mathcal{F}$ is the closure of the convex hull, $\mathrm{co}\,(\mathcal{F}_\Theta)$, it follows that there are sequences of functions $\{f_n\}_{n \ge 1}, \{g_n\}_{n \ge 1} \in \mathrm{co}\,(\mathcal{F}_\Theta)$ such that for every $\epsilon > 0$ and

sufficiently large $n$ it holds $\|f - f_n\|_\infty < \epsilon$ and $\|g - g_n\|_\infty < \epsilon$. Then, for a convex combination of functions $f$ and $g$ and sufficiently large $n$ we have

$$\|\alpha f + (1 - \alpha) g - \alpha f_n - (1 - \alpha) g_n\|_\infty \leq \alpha \|f - f_n\|_\infty + (1 - \alpha) \|g - g_n\|_\infty < \epsilon.$$

From here it follows that for every $0 \leq \alpha \leq 1$ and $f, g \in \mathcal{F}$ it holds $\alpha f + (1 - \alpha) g \in \mathcal{F}$. Thus, we have showed that the hypothesis space $\mathcal{F}$ is a convex set.

As a convex combination of Lipschitz continuous functions is again a Lipschitz continuous function, we have that all functions $f \in \mathrm{co}\,(\mathcal{F}_\Theta)$ are Lipschitz continuous. It remains to prove that all functions from the closure are Lipschitz continuous, as well. Let $f$ and $\{f_n\}_{n \geq 1}$ be defined as above and let $L_\phi$ be the Lipschitz constant of the function $\phi$. We have that it holds

$$\begin{aligned}|f(x) - f(y)| \leq \quad & |f(x) - f_n(x)| + |f_n(x) - f_n(y)| + |f_n(y) - f(y)| < \\ & 2 \|f - f_n\|_\infty + rL_\phi \|x - y\|.\end{aligned}$$

Taking the limit of both sides as $n \to \infty$, we deduce that function $f$ is Lipschitz continuous with a Lipschitz constant bounded by $rL_\phi$.

Depending on the choice of the basis function $\phi$, the hypothesis space can be a space of infinite dimension and the fact that it is bounded and complete does not imply that it is compact, as well. The metric space $(\mathcal{F}, \|\cdot\|_\infty)$ is compact if and only if it is complete and totally bounded [27], i.e., for all $\epsilon > 0$ there exists a finite $\epsilon$-net of $\mathcal{F}$. As the hypothesis space $\mathcal{F}$ is complete by definition, it is sufficient to show that for all $\epsilon > 0$ there exists a finite $\epsilon$-net of $\mathcal{F}$ in $\mathcal{C}(X)$. The set $X$ is a compact subset of finite dimensional Euclidean space and as such it is totally bounded and contained in a centralizable space (see Definition A.4 for details). Then, from Theorem A.4 it follows that

$$\mathcal{N}\left(\mathrm{Lip}_1(X), \epsilon, \|\cdot\|_\infty\right) \leq 2^{\mathcal{N}\left(X, \frac{\epsilon}{2}, \|\cdot\|\right)} \left(2 \left\lceil \frac{2C}{\epsilon} \right\rceil + 1\right),$$

where $\mathrm{Lip}_1(X)$ denotes the set of 1-Lipschitz functions defined on a set $X$, $\mathcal{N}(X, \epsilon, \|\cdot\|)$ denotes the minimal number of points in an $\epsilon$-net of the set $X$ with respect to the metric $\|\cdot\|$, and $C > 0$ is the upper bound on all functions in $\mathrm{Lip}_1(X)$. This result allows us to bound the covering number of the space of Lipschitz continuous functions on the compact set $X$. Namely, from the assumptions about $\mathcal{F}$ we conclude that all functions in $\mathcal{F}$ have Lipschitz constant bounded by $L_\mathcal{F} = rL_\phi$, where $L_\phi$ denotes the Lipschitz constant of the function $\phi$. Then, the upper bound on the covering number of the space $\mathrm{Lip}_{L_\mathcal{F}}(X)$ is given by

$$2^{\mathcal{N}\left(X, \frac{\epsilon}{2L_\mathcal{F}}, \|\cdot\|_2\right)} \left(2 \left\lceil \frac{2r}{\epsilon} \right\rceil + 1\right).$$

Since $\mathcal{F} \subset \mathrm{Lip}_{L_\mathcal{F}}(X)$ and $\mathcal{N}\left(\mathrm{Lip}_{L_\mathcal{F}}(X), \epsilon, \|\cdot\|_\infty\right)$ is finite, the result follows. □

**Theorem 6.** *Let $M > 0$ such that, for all $f \in \mathcal{F}$, $|f(x) - y| \leq M$ almost surely. Then, for all $\epsilon > 0$*

$$\mathbb{P}\left[\mathcal{E}_\rho(f_\mathbf{z}) - \mathcal{E}_\rho(f^*) \leq \epsilon\right] \geq 1 - \mathcal{N}(\mathcal{F}, \epsilon/24M, \|\cdot\|_\infty) \exp\left(-m\epsilon/288M^2\right),$$

*where $f_\mathbf{z}$ and $f^*$ are the minimizers of $\mathcal{E}_\mathbf{z}$ and $\mathcal{E}_\rho$ on the set $\mathcal{F}$, $\mathbf{z} \in Z^m$, and $\mathcal{N}(\mathcal{F}, \epsilon, \|\cdot\|_\infty)$ denotes the $\epsilon$-covering number of $\mathcal{F}$ w.r.t. $\mathcal{C}(X)$.*

*Proof.* The claim follows from Proposition 5 and Theorem A.5. □

**Corollary 7.** *For all $\epsilon > 0$ and all $\delta > 0$, with probability $1 - \delta$, a minimizer of the empirical squared error on the hypothesis space $\mathcal{F}$ is $(\epsilon, \delta)$-consistent when the number of samples $m \in \Omega\left(r(Rs + t)L_\phi \frac{1}{\epsilon^2} + \frac{1}{\epsilon} \ln \frac{1}{\delta}\right)$. Here, $R$ is the radius of a ball containing the set of instances $X$ in its interior, $L_\phi$ is the Lipschitz constant of a function $\phi$, and $r$, $s$, and $t$ are hyperparameters of $\mathcal{F}_\Theta$.*

*Proof.* To derive a sample complexity bound from the corollary we need a tighter bound on the covering number of our hypothesis space than the one provided in Proposition 5. We first give one such bound and then prove the corollary.

The set of instances $X$ is a compact subset of Euclidean space and we can, without the loss of generality, assume that there exists a ball of radius $R$ centered at the origin and containing the set $X$

in its interior. From the definition of the hypothesis space $\mathcal{F}$ we see that the argument of the ridge function $\phi$ is bounded, i.e.,

$$|\langle w, x \rangle + b| \leq \|w\| \, \|x\| + t \leq Rs + t.$$

From here we conclude that the hypothesis space $\mathcal{F}$ is a subset of the space of 1-dimensional Lipschitz continuous functions on the compact interval $[-(Rs + t), Rs + t]$. Then, the covering number of $\mathcal{F}$ is upper bounded by the covering number of the space of $L_{\mathcal{F}}$-Lipschitz continuous one dimensional functions defined on the segment $[-(Rs + t), Rs + t]$.

From Proposition A.7 it follows that the $\epsilon$-covering number of the segment $[-(Rs + t), Rs + t]$ is upper bounded by $\frac{4(Rs+t)}{\epsilon}$. This, together with Theorem A.4 implies that the upper bound on the $\epsilon$-covering number of the hypothesis space $\mathcal{F}$ is given by

$$\mathcal{N}\left(\mathcal{F}, \epsilon, \|\cdot\|_{\infty}\right) \leq 2^{\frac{8r(Rs+t)L_{\phi}}{\epsilon}} \left(2 \left\lceil \frac{2r}{\epsilon} \right\rceil + 1\right). \tag{3}$$

On the other hand, from Theorem 6 we get that for all $\delta > 0$ with probability $1 - \delta$ the empirical estimator is $(\epsilon, \delta)$-consistent when

$$2^{\frac{192r(Rs+t)ML_{\phi}}{\epsilon}} \left(2 \left\lceil \frac{48Mr}{\epsilon} \right\rceil + 1\right) \exp\left(-\frac{m\epsilon}{288M^2}\right) \leq \delta, \text{ or}$$

$$\frac{192r(Rs + t)ML_{\phi}}{\epsilon} \ln 2 + \ln\left(2 \left\lceil \frac{48Mr}{\epsilon} \right\rceil + 1\right) \leq \frac{m\epsilon}{288M^2} - \ln \frac{1}{\delta}.$$

Hence, for all $\epsilon, \delta > 0$ and

$$m \geq \frac{288M^2}{\epsilon} \left[\frac{192r(Rs + t)ML_{\phi}}{\epsilon} \ln 2 + \ln\left(2 \left\lceil \frac{48Mr}{\epsilon} \right\rceil + 1\right) + \ln \frac{1}{\delta}\right] \tag{4}$$

with probability $1 - \delta$ the empirical estimator is $(\epsilon, \delta)$-consistent. $\qquad \square$

**Remark 1.** *The concentration inequality is tighter by a factor of $\frac{1}{\epsilon}$ for convex and compact compared to compact only hypothesis spaces. For instance, this can be seen by comparing the bounds from Theorems A.5 and A.6. In our case with convex and compact hypothesis space $\mathcal{F}$, the final sample complexity bound is still $\Omega\left(\frac{1}{\epsilon^2}\right)$ due to the $\frac{1}{\epsilon}$ factor coming from the $\epsilon$-covering number of $\mathcal{F}$.*

**Remark 2.** *A detailed study of the approximation properties of ridge functions in high dimensional Euclidean spaces is out of the scope for this paper (e.g., one such study can be found in [24]).*

## C  Implementation Details

In this appendix, we provide implementation details for all the considered algorithms – greedy feature construction, á la carte method [31], and random Fourier features approach [26]. As already stated in Section 2.5, the corresponding linear ridge regression optimization problems are solved by casting them as hyperparameter optimization problems [20]. To be as objective as possible to the best performing competing method [31], we have followed the experimental setting outlined there and optimized the hyperparameters with the L-BFGS-B solver from *SciPy*.

### C.1  Greedy Feature Construction

We have implemented a distributed version of Algorithm 2 using a python package *mpi4py*. For the experiments with 100 spectral features the algorithm is simulated using 5 cores on a single physical machine – each core corresponds to one instance of greedy functional descent. The remaining parameters are: the number of data passes $T = 1$, the maximum number of greedy descent steps $p = 20$, precision parameter $\epsilon = 0.01$ that stops the greedy descent when the successive improvement in the accuracy is less than $1\%$, and feature cut-off $\eta$ that is set to $0.0001\%$ of the range of the output variable. For the experiments with 500 spectral features the algorithm is simulated using 5 physical machines. To communicate features more efficiently 5 cores on each of the physical machines is used giving the total number of 25 cores corresponding to 25 instances of greedy functional descent. The remaining parameters for this setting are identical to the ones used in the experiments with 100 features. As the greedy functional descent is stopped when the successive improvement in the accuracy is below $1\%$, the approach terminates sooner than the alternative approaches (w.r.t. the number of constructed features) for simple hypotheses (see Appendix D). In contrast to á la carte [31], we *did not engineer a heuristic for the initial solution* of the hyper-parameter optimization problem. Instead, we have initialized the spectral features by sampling from the standard normal distribution and dividing the entries of the sampled vector with the square root of its dimension.

Having specified the parameter settings for Algorithm 2, we proceed to a discussion regarding the regularization term from the optimization problem defined at line 3 of Algorithm 1. The section concludes with hyperparameter gradients for the cosine-wave feature space introduced in Section 2.2.

**Regularization.** It is frequently the case that generalization properties and the capacity of a hypothesis space are controlled by penalizing the objective function with the squared $l_2$ norm of a parameter vector. For instance, this is the case for the majority of standard activation functions in neural networks literature. The reason behind this choice of the regularizer lies in the fact that these activation functions are monotone and the variation of any such basis function corresponds with the variation in its ridge argument. Assuming that the data is centered, the variation of the ridge argument can be expressed as

$$\int w^\top x x^\top w \rho(x) = w^\top \left( \int x x^\top \rho(x) \right) w = \|w\|^2 .$$

However, if we opt for cosine ridge functions as in Sections 2.2 and 3, then it is not straightforward to relate the smoothness of the basis function to its argument (considered over a given finite sample of the data). Namely, cosine is a periodic function and while spectral parameters with large norms can cause significant variation in the ridge argument, this does not necessarily imply a large variation of the basis function over a finite sample. It is also possible for a parameter vector with the smaller norm to cause more variation in the basis function over a finite sample than the one with the larger norm. We, therefore, opt to regularize the spectrum of the cosine ridge function by penalizing the objective with its squared $\mathcal{L}^2_\rho(X)$ norm. Before we give the regularization term, we first note that the bias term from the cosine-wave features can be eliminated using the trigonometric additive formulas and then the cosine-wave basis function takes the from

$$\phi_{w,a}(x) = a_1 \sin\left(w^\top x\right) + a_2 \cos\left(w^\top x\right) . \tag{5}$$

Now, taking the squared $\mathcal{L}^2_\rho(X)$ norm of this function we get

$$
\begin{aligned}
\|\phi_{w,a}\|^2_\rho \;&= a_1^2 \int \sin^2\left(w^\top x\right)\rho(x) + a_2^2 \int \cos^2\left(w^\top x\right)\rho(x) + a_1 a_2 \int \sin\left(2w^\top x\right)\rho(x) \\
&= \frac{a_1^2 + a_2^2}{2} + \frac{a_2^2 - a_1^2}{2} \int \cos\left(2r\right)\mu_w(r) + a_1 a_2 \int \sin\left(2r\right)\mu_w(r) ,
\end{aligned}
$$

where $\mu_w(r) = \int \rho(x \mid w^\top x = r)$. If we assume that the probability measure $\rho$ is symmetric, then we have that $\mu_w(r) = \mu_w(-r)$ and using the fact that $\sin(2r)$ is an odd function, we obtain $\int \sin(2w^\top x) \rho(x) = 0$. In the absence of the marginal distribution $\rho$, the integral $\int \cos(2r) \mu_w(r)$ can be estimated from the training sample with $\frac{1}{m} \sum_{i=1}^{m} \cos(2w^\top x_i)$, where $x_i \overset{\text{i.i.d.}}{\sim} \rho(x)$.

**Hyper-parameter optimization.** We now formulate the optimization problem (line 3, Algorithm 1) for the setting with cosine-wave features and provide the gradients for all the hyperparameters. The optimization problem can be specified as

$$\min \quad \frac{1}{m} \sum_{i=1}^{m} \left( c_0 f_{0,i} + c_1 \sin(w^\top x_i) + c_2 \cos(w^\top x_i) - y_i \right)^2 +$$

$$\lambda \left( \frac{c_0^2}{m} \sum_{i=1}^{m} f_{0,i}^2 + \frac{c_1^2 + c_2^2}{2} + \frac{c_2^2 - c_1^2}{2m} \sum_{i=1}^{m} \cos(2w^\top x_i) + \frac{2c_0 c_1}{m} \sum_{i=1}^{m} \sin(w^\top x_i) f_{0,i} + \right.$$

$$\left. \frac{2c_0 c_2}{m} \sum_{i=1}^{m} \cos(w^\top x_i) f_{0,i} + \frac{c_1 c_2}{m} \sum_{i=1}^{m} \sin(2w^\top x_i) \right),$$

where $w$ and $\lambda$ are optimized as hyperparameters and amplitude vector $c$ as a regressor. As the regressor is completely determined by the choice of $\lambda$, it is sufficient to optimize this problem only by the hyperparameters $w$ and $\lambda$. We want to choose these parameters via $k$-fold cross-validation and in order to achieve this we follow the procedure proposed by Keerthi et al. [20]. Let us denote the above described 3-dimensional feature representation of the data with $Z_w \in \mathbb{R}^{m \times 3}$ and set $\sigma_0 = \frac{1}{m} \sum_{i=1}^{m} f_{0,i}^2$, $\sigma_1 = \frac{1}{m} \sum_{i=1}^{m} \sin(w^\top x_i) f_{0,i}$, $\sigma_2 = \frac{1}{m} \sum_{i=1}^{m} \cos(w^\top x_i) f_{0,i}$, $\sigma_3 = \frac{1}{m} \sum_{i=1}^{m} \sin(2w^\top x_i)$, $\sigma_4 = \frac{1}{m} \sum_{i=1}^{m} \cos(2w^\top x_i)$. Now, in the place of the identity matrix in the derivative of the ridge regression objective function we have the matrix

$$D = \begin{bmatrix} \sigma_0 & \sigma_1 & \sigma_2 \\ \sigma_1 & 0.5(1 - \sigma_4) & 0.5\sigma_3 \\ \sigma_2 & 0.5\sigma_3 & 0.5(1 + \sigma_4) \end{bmatrix}$$

At this point our derivation follows closely the derivation by Keerthi et al. [20]. Taking the derivatives with respect to $c$ and setting the gradient of the loss to zero we get

$$Z_w^\top Z_w c - Z_w^\top y + m\lambda D c = 0,$$

$$\left( Z_w^\top Z_w + m\lambda D \right) c = Z_w^\top y.$$

Let us now denote with $P = Z_w^\top Z_w + m\lambda D$, $q = Z_w^\top y$, and $\theta = (w, \lambda)$. We note here that $P$ and $q$ are defined over the *training instances* $x$ and their labels $y$. We now take the implicit derivative of this equation to obtain the derivative of the regressor $c$ with respect to the hyperparameters, i.e.,

$$\frac{\partial c}{\partial \theta} = P^{-1} \left( \frac{\partial q}{\partial \theta} - \frac{\partial P}{\partial \theta} c \right).$$

As already stated, the choice of $\lambda$ directly determines the coefficients $c$ and to obtain these we need to perform the hyperparameter selection which is done over the validation samples. In other words,

$$\theta^* = \arg\min_\theta \frac{1}{k} \sum_{i=1}^{k} \frac{1}{|F_i|} \sum_{(x,y) \in F_i} \left( c^\top z_w(x) - y \right)^2,$$

where $F_i$ denotes one of $k$ validation folds in the $k$-fold cross-validation and $z_w(x)$ is the 3-dimensional representation of an instance $x \in X$. Let us now consider only the sample from one validation fold and denote it with $F$. At the same time let $F^c$ denotes its complement or the training sample when $F$ is used as the validation fold. Here, we note that $(x, y) \in F$ are *different* from samples participating in the definitions of $P$ and $q$ when taking $F$ as the validation fold. Taking derivatives with respect to $\theta$ we get the hyperparameter gradient

$$\frac{2}{|F|} \sum_{(x,y) \in F} \left( c^\top z_w(x) - y \right) \left( \frac{\partial z_w(x)}{\partial \theta} c + z_w(x) P^{-1} \left( \frac{\partial q}{\partial \theta} - \frac{\partial P}{\partial \theta} c \right) \right).$$

Let us introduce a vector $t = (t_0, t_1, t_2)$ as a solution to the following 3-dimensional linear system

$$Pt = \frac{1}{|F|} \sum_{(x,y) \in F} \left( c^\top z_w(x) - y \right) z_w(x).$$

We then write the derivative of each term in the hyperparameter gradient separately as

$$\frac{\partial}{\partial w} \left( c^\top z_w(x) \right) = \left( c_1 \cos \left( w^\top x \right) - c_2 \sin \left( w^\top x \right) \right) x$$

$$\frac{\partial}{\partial w} \left( t^\top q \right) = \sum_{(x,y) \in F^c} \left( t_1 \cos \left( w^\top x \right) - t_2 \sin \left( w^\top x \right) \right) xy$$

$$\frac{\partial}{\partial w} \left( t^\top P c \right) = (1 + \lambda) \left( t_0 c_1 + t_1 c_0 \right) \sum_{i=1}^{|F^c|} f_{0,i} \cos \left( w^\top x_i \right) x_i -$$

$$(1 + \lambda) \left( t_0 c_2 + t_2 c_0 \right) \sum_{i=1}^{|F^c|} f_{0,i} \sin \left( w^\top x_i \right) x_i + (1 + \lambda) \left( t_1 c_2 + t_2 c_1 \right) \sum_{x \in F^c} \cos \left( 2 w^\top x \right) x +$$

$$(1 + \lambda) \left( t_1 c_1 - t_2 c_2 \right) \sum_{x \in F^c} \sin \left( 2 w^\top x \right) x$$

$$\frac{\partial}{\partial \lambda} = m \, t^\top D c$$

Performing the gradient descent using these hyperparameter gradients we obtain both the spectrum $w$ and the amplitudes $c$. The spectrum regularization term which is defined using the empirical estimates of the sine and cosine integrals affects the gradient with respect to $w$ via the $\lambda$ factor in the third expression. In our experiments, we have observed that the capacity parameter $\lambda$ usually takes the value below $10^{-4}$. Thus, the influence of the spectrum regularization term is less significant than the amplitude regularization term. For this reason, in our implementation we only penalize the empirical squared error objective with the squared norm of the amplitude vector, i.e., $\Omega(c, w) = \|c\|_2^2$.

### C.2 Random Fourier features

As already pointed out in Theorem A.1 and Appendix A, any shift-invariant positive definite kernel can be represented as a Fourier transform of a positive measure. Thus, in order to generate a kernel feature map it is sufficient to sample spectral frequencies from this measure. Genton [17] and Rahimi and Recht [26] have provided the parameterized spectral density functions corresponding to Gaussian, Laplace, and Cauchy kernels. We use these parameterizations to generate spectral features and then train a linear ridge regression model in the constructed feature space. To choose the most suitable parameterization, we cross-validate 10 parameters from the log-space of $[-3, 2]$.

### C.3 A la carte

The random Fourier features approach [26] is an efficient method for the approximation of functions from shift-invariant reproducing kernel Hilbert spaces. However, this method requires an *a priori* specification of a suitable spectral measure which is often not feasible. To address this shortcoming, Yang et al. [31] estimate a data-dependent spectral distribution using a mixture of Gaussians and represent the regression estimator as

$$f(x) = \sum_{j=1}^{n} \alpha_j \sin \left( w_j^\top x \right) + \alpha_j^{'} \cos \left( w_j^\top x \right),$$

where $n$ denotes the number of spectral features, and

$$w \sim \sum_{k=1}^{Q} \frac{\gamma_k}{\sqrt{(2\pi)^d |\Sigma_k|}} \exp \left( -\frac{(x - \mu_k)^\top \Sigma_k^{-1} (x - \mu_k)}{2} \right)$$

with $\Sigma_k$ diagonal, $\gamma \geq 0$, and $\sum_{k=1}^{Q} \gamma_k = 1$. The proposed algorithm finds a feature representation together with a linear model by optimizing the marginal likelihood of a Gaussian process. As we have chosen to compare all the feature construction approaches using the standard linear regression, we provide an equivalent implementation of this approach based on the hyper-parameter optimization method proposed by Keerthi et al. [20]. To make the comparison as objective as possible, we have parallelized the implementation of this algorithm and simulated it by following the ARD-heuristic for choosing the initial solution [31, Supplementary material]. In all the experimental settings (with 100 and 500 features), we have run this algorithm using $Q = 1$, $Q = 2$, and $Q = 5$ mixture components.

**Optimization problem.** We first give the optimization objective for the non-Gaussian process case,

$$
\min \quad \frac{1}{m} \sum_{i=1}^{m} \left[ \sum_{q=1}^{Q} \nu_q^2 \sum_{j=1}^{s} \alpha_{qj} \sin \left( u_{qj}^\top \Sigma_q^{1/2} x_i + \mu_q^\top x_i \right) + \beta_{qj} \cos \left( u_{qj}^\top \Sigma_q^{1/2} x_i + \mu_q^\top x_i \right) - y_i \right]^2 +
$$
$$
\lambda \left( \|\alpha\|^2 + \|\beta\|^2 \right),
$$

where $\alpha$ and $\beta$ are optimized as regressors and $\mu_q$, $\Sigma_q$ (diagonal covariance matrix), and $\lambda$ as hyper-parameters. The $u$-vectors are random vectors sampled from the multivariate standard normal distribution. These vectors act as a regularization term on the spectrum of the cosine features forcing the frequencies to stay in the pre-specified number of clusters/components.

**Hyper-parameter optimization.** Let us denote $\Sigma_q^{1/2}$ with $D_q$, parameterized features with $Z_\theta \in \mathbb{R}^{m \times Qs}$, hyperparameters with $\theta = (\mu, D, \nu)$, and regressors with $c = (\alpha, \beta)$. Similar to the previous section set $P = Z_\theta^\top Z_\theta + m\lambda \mathbb{I}$ and $q = Z_\theta^\top y$. Following the same principles for the implicit derivation, we obtain the gradient terms of the hyper-parameter objective function:

$$
\frac{\partial}{\partial \mu_q} \left( c^\top z_\theta(x) \right) = \nu_q^2 \left( \alpha_q^\top \cos \left( U_q D_q x \oplus \mu_q^\top x \right) - \beta_q^\top \sin \left( U_q D_q x \oplus \mu_q^\top x \right) \right) x,
$$

$$
\frac{\partial}{\partial D_q} \left( c^\top z_\theta(x) \right) = \nu_q^2 \left( (\alpha_q \odot U_q)^\top \cos \left( U_q D_q x \oplus \mu_q^\top x \right) - (\beta_q \odot U_q)^\top \sin \left( U_q D_q x \oplus \mu_q^\top x \right) \right) \odot x,
$$

$$
\frac{\partial}{\partial \nu_q} \left( c^\top z_\theta(x) \right) = 2\nu_q \left( \alpha_q^\top \sin \left( U_q D_q x \oplus \mu_q^\top x \right) + \beta_q^\top \cos \left( U_q D_q x \oplus \mu_q^\top x \right) \right),
$$

$$
\frac{\partial}{\partial \mu_q} \left( t^\top q \right) = \sum_{(x,y) \in F^c} y \nu_q^2 \left( t_{q\alpha}^\top \cos \left( U_q D_q x \oplus \mu_q^\top x \right) - t_{q\beta}^\top \sin \left( U_q D_q x \oplus \mu_q^\top x \right) \right) x,
$$

$$
\frac{\partial}{\partial D_q} \left( t^\top q \right) = \sum_{(x,y) \in F^c} y \nu_q^2 \left( (t_{q\alpha} \odot U_q)^\top \cos \left( U_q D_q x \oplus \mu_q^\top x \right) - (t_{q\beta} \odot U_q)^\top \sin \left( U_q D_q x \oplus \mu_q^\top x \right) \right) \odot x,
$$

$$
\frac{\partial}{\partial \nu_q} \left( t^\top q \right) = 2\nu_q \sum_{(x,y) \in F^c} t_{q\alpha}^\top \sin \left( U_q D_q x \oplus \mu_q^\top x \right) + t_{q\beta}^\top \cos \left( U_q D_q x \oplus \mu_q^\top x \right),
$$

$$
\frac{\partial}{\partial \mu_q} \left( t^\top P c \right) = \sum_{(x,y) \in F^c} \nu_q^4 \Bigg\{
$$
$$
\left[ t_\alpha^\top \sin \left( U D x \oplus \mu^\top x \right) + t_\beta^\top \cos \left( U D x \oplus \mu^\top x \right) \right] \cdot \left[ \alpha_q^\top \cos \left( U_q D_q x \oplus \mu_q^\top x \right) - \beta_q^\top \sin \left( U_q D_q x \oplus \mu_q^\top x \right) \right] +
$$
$$
\left[ \alpha^\top \sin \left( U D x \oplus \mu^\top x \right) + \beta^\top \cos \left( U D x \oplus \mu^\top x \right) \right] \cdot \left[ t_{q\alpha}^\top \cos \left( U_q D_q x \oplus \mu_q^\top x \right) - t_{q\beta}^\top \sin \left( U_q D_q x \oplus \mu_q^\top x \right) \right] \Bigg\} x,
$$

$$\frac{\partial}{\partial \nu_q}\left(t^\top Pc\right) = 2 \sum_{(x,y)\in F^c} \nu_q^3 \Bigg\{$$

$$\left[t_\alpha^\top \sin\left(UDx \oplus \mu^\top x\right) + t_\beta^\top \cos\left(UDx \oplus \mu^\top x\right)\right] \cdot \left[\alpha^\top \sin\left(UDx \oplus \mu^\top x\right) + \beta^\top \cos\left(UDx \oplus \mu^\top x\right)\right]\Bigg\}$$

$$\frac{\partial}{\partial D_q}\left(t^\top Pc\right) = \sum_{(x,y)\in F^c} \nu_q^4 \Bigg\{$$

$$\left[t_\alpha^\top \sin\left(UDx \oplus \mu^\top x\right) + t_\beta^\top \cos\left(UDx \oplus \mu^\top x\right)\right] \cdot \left[(\alpha_q \odot U_q)^\top \cos\left(U_q D_q x \oplus \mu_q^\top x\right) -\right.$$

$$\left.(\beta_q \odot U_q)^\top \sin\left(U_q D_q x \oplus \mu_q^\top x\right)\right] +$$

$$\left[\alpha^\top \sin\left(UDx \oplus \mu^\top x\right) + \beta^\top \cos\left(UDx \oplus \mu^\top x\right)\right] \cdot \left[(t_{q\alpha} \odot U_q)^\top \cos\left(U_q D_q x \oplus \mu_q^\top x\right) -\right.$$

$$\left.(t_{q\beta} \odot U_q)^\top \sin\left(U_q D_q x \oplus \mu_q^\top x\right)\right]\Bigg\} \odot x$$

$$\frac{\partial}{\partial \lambda} = m\, t^\top c,$$

where $\oplus$ and $\odot$ denote element-wise addition and multiplication, $U \in \mathbb{R}^{Qs \times d}$ and it consists of blocks $U_q \in \mathbb{R}^{s \times d}$ such that each block contains row vectors sampled from a multivariate standard normal distribution.

The cost of computing the gradient of hyperparameters for á la carte involves solving an $n = Qs$ dimensional linear system. This system needs to be solved for each validation fold in the $k$-fold splitting, required for the optimization of the hyperparameters over the validation samples. As this can be computationally intensive on a single core, we have parallelized our implementation of á la carte by computing the parts of hyperparameter gradient that correspond to different validation folds on different cores. For the inner cross-validation performed with 5-fold splitting this has resulted in a speed up of approximately 4-5 times compared to a single core implementation. In Section 3 and Appendix D.1 we report the walltimes of the parallelized implementation of á la carte.

**Initial solution.** As already stated, we follow the instructions from the supplementary material of Yang et al. [31] and initialize the means to vectors that are close to zero. The $\nu$ parameters are initialized by setting their values to the standard deviation of outputs divided by the number of components $Q$. The diagonal covariance matrices are initialized by following the ARD-heuristic. As reported in [31], we simulate the algorithm with 10 random restarts such that for each initial solution the algorithm makes 20 iterations of L-BFGS-B minimization and then continues with the best hyperparameter vector for another 200 iterations.

# D  Results

## D.1  Á la Carte

Table 2: The mean and standard deviation of the root mean squared error are computed after performing 10-fold cross-validation. The fold splitting is done such that all algorithms train and predict over identical samples. The reported walltime is the average time it takes a method to cross-validate one fold.

| DATASET | $m$ | $d$ | $n = 100$ | | | | | |
|---|---|---|---|---|---|---|---|---|
| | | | $Q = 1, s = 100$ | | $Q = 2, s = 50$ | | $Q = 5, s = 20$ | |
| | | | ERROR | WALLTIME | ERROR | WALLTIME | ERROR | WALLTIME |
| parkinsons tm (total) | 5875 | 21 | 0.81 ($\pm$0.67) | 00 : 07 : 58 | 0.73 ($\pm$0.33) | 00 : 08 : 29 | 0.78 ($\pm$0.13) | 00 : 05 : 19 |
| ujindoorloc (latitude) | 21048 | 527 | 6.21 ($\pm$0.41) | 00 : 27 : 41 | 6.94 ($\pm$0.66) | 00 : 45 : 55 | 6.19 ($\pm$0.76) | 01 : 21 : 58 |
| ct-slice | 53500 | 380 | 4.11 ($\pm$0.25) | 00 : 46 : 56 | 3.86 ($\pm$0.28) | 01 : 18 : 00 | 3.82 ($\pm$0.64) | 03 : 31 : 25 |
| Year Prediction MSD | 515345 | 90 | 10.10 ($\pm$0.07) | 02 : 49 : 21 | 10.03 ($\pm$0.08) | 02 : 32 : 09 | 9.94 ($\pm$0.08) | 05 : 29 : 14 |
| delta-ailerons | 7129 | 5 | 3.83 ($\pm$0.18) | 00 : 04 : 19 | 3.84 ($\pm$0.27) | 00 : 05 : 27 | 3.73 ($\pm$0.20) | 00 : 05 : 13 |
| kinematics | 8192 | 8 | 6.21 ($\pm$0.54) | 00 : 10 : 21 | 5.31 ($\pm$0.34) | 00 : 09 : 30 | 5.03 ($\pm$0.23) | 00 : 11 : 28 |
| cpu-activity | 8192 | 21 | 2.59 ($\pm$0.17) | 00 : 08 : 22 | 2.77 ($\pm$0.33) | 00 : 06 : 19 | 2.68 ($\pm$0.27) | 00 : 09 : 24 |
| bank | 8192 | 32 | 9.72 ($\pm$0.32) | 00 : 12 : 03 | 9.79 ($\pm$0.29) | 00 : 10 : 27 | 9.84 ($\pm$0.30) | 00 : 12 : 48 |
| pumadyn | 8192 | 32 | 3.17 ($\pm$0.07) | 00 : 10 : 34 | 3.18 ($\pm$0.06) | 00 : 11 : 01 | 3.24 ($\pm$0.07) | 00 : 13 : 17 |
| delta-elevators | 9517 | 6 | 5.28 ($\pm$0.17) | 00 : 03 : 31 | 5.27 ($\pm$0.17) | 00 : 06 : 52 | 5.28 ($\pm$0.18) | 00 : 07 : 07 |
| ailerons | 13750 | 40 | 4.62 ($\pm$0.34) | 00 : 08 : 42 | 4.57 ($\pm$0.12) | 00 : 09 : 54 | 4.89 ($\pm$0.43) | 00 : 16 : 34 |
| pole-telecom | 15000 | 26 | 8.73 ($\pm$0.52) | 00 : 12 : 39 | 7.34 ($\pm$0.32) | 00 : 15 : 00 | 7.16 ($\pm$0.55) | 00 : 20 : 34 |
| elevators | 16599 | 18 | 3.46 ($\pm$0.23) | 00 : 07 : 51 | 3.70 ($\pm$0.55) | 00 : 07 : 41 | 3.37 ($\pm$0.55) | 00 : 21 : 20 |
| cal-housing | 20640 | 8 | 13.61 ($\pm$0.35) | 00 : 09 : 49 | 13.07 ($\pm$1.53) | 00 : 12 : 17 | 12.69 ($\pm$0.47) | 00 : 11 : 14 |
| breiman | 40768 | 10 | 4.01 ($\pm$0.03) | 00 : 12 : 34 | 4.02 ($\pm$0.04) | 00 : 09 : 13 | 4.06 ($\pm$0.04) | 00 : 13 : 52 |
| friedman | 40768 | 10 | 3.16 ($\pm$0.03) | 00 : 18 : 58 | 3.16 ($\pm$0.03) | 00 : 19 : 46 | 3.37 ($\pm$0.46) | 00 : 18 : 43 |

Table 3: The mean and standard deviation of the root mean squared error are computed after performing 10-fold cross-validation. The fold splitting is done such that all algorithms train and predict over identical samples. The reported walltime is the average time it takes a method to cross-validate one fold.

| DATASET | $m$ | $d$ | $n = 500$ | | | | | |
|---|---|---|---|---|---|---|---|---|
| | | | $Q = 1, s = 500$ | | $Q = 2, s = 250$ | | $Q = 5, s = 100$ | |
| | | | ERROR | WALLTIME | ERROR | WALLTIME | ERROR | WALLTIME |
| parkinsons tm (total) | 5875 | 21 | 0.29 ($\pm$0.33) | 00 : 30 : 00 | 0.34 ($\pm$0.17) | 00 : 37 : 05 | 0.31 ($\pm$0.17) | 00 : 27 : 15 |
| ujindoorloc (latitude) | 21048 | 527 | 8.08 ($\pm$1.67) | 01 : 34 : 01 | 7.83 ($\pm$1.05) | 02 : 02 : 19 | 6.99 ($\pm$0.97) | 02 : 23 : 15 |
| ct-slice | 53500 | 380 | 2.98 ($\pm$0.07) | 02 : 43 : 28 | 2.97 ($\pm$0.19) | 04 : 09 : 43 | 2.73 ($\pm$0.29) | 06 : 11 : 12 |
| Year Prediction MSD | 515345 | 90 | 10.00 ($\pm$0.07) | 07 : 51 : 20 | 9.94 ($\pm$0.07) | 08 : 55 : 38 | 9.92 ($\pm$0.07) | 11 : 58 : 41 |
| delta-ailerons | 7129 | 5 | 3.82 ($\pm$0.18) | 00 : 14 : 37 | 3.85 ($\pm$0.37) | 00 : 18 : 23 | 3.73 ($\pm$0.24) | 00 : 25 : 14 |
| kinematics | 8192 | 8 | 5.34 ($\pm$0.48) | 00 : 29 : 45 | 4.82 ($\pm$0.32) | 00 : 41 : 13 | 5.01 ($\pm$0.76) | 00 : 38 : 53 |
| cpu-activity | 8192 | 21 | 2.47 ($\pm$0.36) | 00 : 52 : 16 | 2.52 ($\pm$0.20) | 00 : 29 : 34 | 2.62 ($\pm$0.15) | 00 : 25 : 13 |
| bank | 8192 | 32 | 9.62 ($\pm$0.29) | 00 : 51 : 08 | 9.97 ($\pm$0.37) | 00 : 48 : 22 | 9.87 ($\pm$0.42) | 00 : 49 : 48 |
| pumadyn | 8192 | 32 | 3.12 ($\pm$0.07) | 00 : 44 : 17 | 3.17 ($\pm$0.05) | 00 : 44 : 28 | 3.42 ($\pm$0.15) | 00 : 57 : 33 |
| delta-elevators | 9517 | 6 | 5.27 ($\pm$0.18) | 00 : 15 : 59 | 5.28 ($\pm$0.18) | 00 : 22 : 44 | 5.23 ($\pm$0.18) | 00 : 32 : 30 |
| ailerons | 13750 | 40 | 4.50 ($\pm$0.10) | 00 : 41 : 45 | 4.49 ($\pm$0.17) | 00 : 36 : 54 | 4.77 ($\pm$0.40) | 01 : 05 : 07 |
| pole-telecom | 15000 | 26 | 6.30 ($\pm$0.45) | 01 : 08 : 32 | 5.35 ($\pm$0.27) | 01 : 17 : 48 | 5.20 ($\pm$0.51) | 01 : 39 : 22 |
| elevators | 16599 | 18 | 3.28 ($\pm$0.27) | 01 : 01 : 44 | 3.37 ($\pm$0.12) | 00 : 36 : 30 | 3.13 ($\pm$0.24) | 01 : 20 : 58 |
| cal-housing | 20640 | 8 | 12.27 ($\pm$1.51) | 01 : 03 : 49 | 12.15 ($\pm$0.43) | 00 : 55 : 06 | 12.70 ($\pm$1.01) | 01 : 01 : 37 |
| breiman | 40768 | 10 | 4.01 ($\pm$0.04) | 00 : 39 : 36 | 4.02 ($\pm$0.04) | 00 : 35 : 45 | 4.03 ($\pm$0.03) | 01 : 04 : 16 |
| friedman | 40768 | 10 | 3.16 ($\pm$0.04) | 00 : 55 : 19 | 3.24 ($\pm$0.06) | 00 : 56 : 33 | 3.25 ($\pm$0.09) | 01 : 39 : 37 |

## D.2  Random Fourier Features

Table 4: The mean and standard deviation of the root mean squared error are computed after performing 10-fold cross-validation. The fold splitting is done such that all algorithms train and predict over identical samples. The reported walltime is the average time it takes a method to cross-validate one fold.

| DATASET | $m$ | $d$ | $n = 100$ | | | | $n = 500$ | | | |
|---|---|---|---|---|---|---|---|---|---|---|
| | | | GAUSS | CAUCHY | LAPLACE | WALLTIME | GAUSS | CAUCHY | LAPLACE | WALLTIME |
| parkinsons tm | 5875 | 21 | 5.81 ($\pm$0.32) | 5.79 ($\pm$0.41) | 6.22 ($\pm$1.31) | 00 : 04 : 38 | 4.75 ($\pm$0.70) | 4.63 ($\pm$0.37) | 4.34 ($\pm$0.22) | 00 : 11 : 10 |
| ujindoorloc | 21048 | 527 | 12.55 ($\pm$0.60) | 12.36 ($\pm$0.67) | 10.23 ($\pm$0.88) | 00 : 05 : 02 | 7.40 ($\pm$0.25) | 7.19 ($\pm$0.31) | 5.53 ($\pm$0.48) | 00 : 24 : 41 |
| ct-slice | 53500 | 380 | 11.45 ($\pm$0.30) | 11.44 ($\pm$0.31) | 11.69 ($\pm$0.50) | 00 : 04 : 32 | 7.85 ($\pm$0.17) | 7.77 ($\pm$0.09) | 7.90 ($\pm$0.13) | 00 : 49 : 10 |
| Year Prediction | 515345 | 90 | 10.75 ($\pm$0.04) | 10.79 ($\pm$0.34) | 11.07 ($\pm$0.13) | 00 : 10 : 26 | 10.53 ($\pm$0.04) | 10.51 ($\pm$0.03) | 10.46 ($\pm$0.06) | 03 : 10 : 41 |
| delta-ailerons | 7129 | 5 | 3.84 ($\pm$0.14) | 3.84 ($\pm$0.14) | 3.86 ($\pm$0.14) | 00 : 04 : 02 | 3.82 ($\pm$0.13) | 3.84 ($\pm$0.15) | 3.81 ($\pm$0.15) | 00 : 15 : 35 |
| kinematics | 8192 | 8 | 11.09 ($\pm$0.26) | 11.01 ($\pm$0.25) | 11.47 ($\pm$0.39) | 00 : 03 : 27 | 7.33 ($\pm$0.53) | 7.37 ($\pm$0.43) | 8.17 ($\pm$0.31) | 00 : 11 : 53 |
| cpu-activity | 8192 | 21 | 6.72 ($\pm$0.62) | 5.94 ($\pm$0.59) | 3.90 ($\pm$0.66) | 00 : 04 : 38 | 3.10 ($\pm$0.17) | 3.05 ($\pm$0.17) | 2.75 ($\pm$0.25) | 00 : 14 : 31 |
| bank | 8192 | 32 | 10.15 ($\pm$0.46) | 10.13 ($\pm$0.42) | 10.10 ($\pm$0.46) | 00 : 04 : 38 | 9.91 ($\pm$0.44) | 9.97 ($\pm$0.49) | 9.92 ($\pm$0.45) | 00 : 15 : 53 |
| pumadyn | 8192 | 32 | 15.19 ($\pm$0.29) | 15.18 ($\pm$0.29) | 15.20 ($\pm$0.29) | 00 : 04 : 50 | 15.20 ($\pm$0.28) | 15.18 ($\pm$0.31) | 15.25 ($\pm$0.27) | 00 : 16 : 52 |
| delta-elevators | 9517 | 6 | 5.30 ($\pm$0.14) | 5.30 ($\pm$0.13) | 5.28 ($\pm$0.13) | 00 : 05 : 16 | 5.29 ($\pm$0.15) | 5.27 ($\pm$0.14) | 5.27 ($\pm$0.14) | 00 : 14 : 49 |
| ailerons | 13750 | 40 | 4.77 ($\pm$0.16) | 4.77 ($\pm$0.21) | 4.89 ($\pm$0.07) | 00 : 03 : 58 | 4.53 ($\pm$0.11) | 4.52 ($\pm$0.10) | 4.58 ($\pm$0.12) | 00 : 17 : 40 |
| pole-telecom | 15000 | 26 | 24.26 ($\pm$0.75) | 22.62 ($\pm$0.68) | 25.07 ($\pm$1.37) | 00 : 04 : 44 | 18.08 ($\pm$0.56) | 17.53 ($\pm$0.46) | 15.63 ($\pm$0.63) | 00 : 19 : 08 |
| elevators | 16599 | 18 | 4.11 ($\pm$0.23) | 3.88 ($\pm$0.21) | 4.09 ($\pm$0.54) | 00 : 04 : 43 | 3.44 ($\pm$0.19) | 3.56 ($\pm$0.37) | 3.39 ($\pm$0.15) | 00 : 19 : 24 |
| cal-housing | 20640 | 8 | 12.99 ($\pm$0.36) | 12.66 ($\pm$0.35) | 12.83 ($\pm$0.53) | 00 : 05 : 27 | 11.78 ($\pm$0.38) | 11.80 ($\pm$0.43) | 11.51 ($\pm$0.37) | 00 : 19 : 02 |
| breiman | 40768 | 10 | 4.01 ($\pm$0.03) | 4.01 ($\pm$0.03) | 4.02 ($\pm$0.03) | 00 : 04 : 26 | 4.01 ($\pm$0.03) | 4.01 ($\pm$0.03) | 4.01 ($\pm$0.03) | 00 : 24 : 45 |
| friedman | 40768 | 10 | 5.15 ($\pm$0.10) | 5.25 ($\pm$0.16) | 5.06 ($\pm$0.32) | 00 : 03 : 58 | 3.30 ($\pm$0.03) | 3.29 ($\pm$0.03) | 3.26 ($\pm$0.04) | 00 : 21 : 17 |