[Reviews · NeurIPS 2016]

Reviewer 1

Summary

The work describes a greedy feature construction approach for linear models. The main idea is to extend functional gradient descent by incrementally constructing the feature space during the iterative process, i.e., for each descent step, an additional feature is added that approximates well the residual function. The overall approach is motivated in Section 1. Section 2 provides some basic preliminaries, followed by a description and analysis of the new greedy feature construction approach given in Section 3. The final algorithm is given in Section 4 (sequential and parallel version), followed by an experimental evaluation in Section 5 and a discussion provided in Section 6. The supplemental material contains additional preliminaries, various proofs, and additional implementation details and experimental results.

Qualitative Assessment

The paper is well written and technically sound. I like the idea of coupling the construction of the feature space with the underlying optimization process. The theoretical derivations provide a convergence rate for the greedy approach as well as generalization bounds. The experimental evaluation is carefully conducted and demonstrates the benefits of the proposed method over its competitors (in particular, over the "à la carte method"). Related work is discussed in detail in Section 6. I think the work provides some novel contributions. In particular, the theoretical derivations seem to be original to me. The approach might also have some practical impact in future (linear models along with feature construction/selection are often used in various domains). However, the theoretical part of the work (Section 3) does not really belong to my area of expertise, so I cannot fully judge the potential of the contributions made (low confidence). Minor comments: - Subsection 4.1 is the only subsection in Section 4., maybe remove this separation? - Section 4, line 209: "Each machines for parallel ..." - Maybe: Move the runtime analysis of the approach from Section 6 to Section 4

Confidence in this Review

1-Less confident (might not have understood significant parts)


Reviewer 2

Summary

In this paper, the authors propose to model the response variable as a (potentially) nonlinear function of the form a f(w’x + b), where f() is an appropriately chosen function. They propose a procedure to learn the function by fitting the residual at each step of an iterative procedure. Rigorous guarantees are provided for the existence of an optimum, and experiments on several datasets demonstrate the applicability of their method.

Qualitative Assessment

The paper is well written, and easy to understand. most of my comments are minor, and are provided below. My biggest concern is a lack of comparison to other related approaches, such as single index models. It seems to be that the method you propose is somewhere between SIM and kernel methods, and while you do better than the latter, the former might be better (in terms of error, but slower algorithmically or need more data). So a comparison is warranted - Line 17: here and elsewhere, you use the term “capacity”. Can you make the notion of capacity precise? Does it relate to how dense the space of relevant functions is? - Line 31: “hypothesis” —> hypotheses - Should the last line in Definition 1 hold for all n? - Definition 1 : should the inf be over both h and g? Or can h be any arbitrary function in co(fn,g) + \eps ? Or is it \forall h? - Definition 2: is B* the dual space? This has not been defined previously. - Table 1: Please also compare to the random kitchen sinks method. You mention that a la carte is better, but a comparison would be nice. - How does your work compare to the (vast) literature on learning Single Index Models? Please see [1- 3] and references therein, and please compare to those methods as well. [1] Kalai, Adam Tauman, and Ravi Sastry. "The Isotron Algorithm: High-Dimensional Isotonic Regression." COLT. 2009. [2] Ichimura, Hidehiko. "Semiparametric least squares (SLS) and weighted SLS estimation of single-index models." Journal of Econometrics 58.1 (1993): 71-120. [3] Kakade, Sham M., et al. "Efficient learning of generalized linear and single index models with isotonic regression." Advances in Neural Information Processing Systems. 2011.

Confidence in this Review

2-Confident (read it all; understood it all reasonably well)


Reviewer 3

Summary

This paper presents an effective method for feature construction by exploiting the scalability of existing algorithms for training linear models while overcoming their low capacity on input features. The authors first embed the data into a space with a high capacity set of linear hypotheses, and then use a greedy approach to construct features by fitting residuals. Overall, I think the contribution of this paper is significant. As noted by the authors, linear models are frequently used for learning on large scale data sets. Yet, the set of linear hypotheses defined on input features is usually of low capacity, thus leading to inaccurate descriptions of target concepts. By embedding the input data into a high capacity set of linear hypotheses and then constructing features in a greedy manner, the proposed method can construct a high capacity feature space. The experimental results also demonstrate the effectiveness of the proposed approach.

Qualitative Assessment

This paper presents an effective method for feature construction by exploiting the scalability of existing algorithms for training linear models while overcoming their low capacity on input features. The authors first embed the data into a space with a high capacity set of linear hypotheses, and then use a greedy approach to construct features by fitting residuals. Overall, I think the contribution of this paper is significant. As noted by the authors, linear models are frequently used for learning on large scale data sets. Yet, the set of linear hypotheses defined on input features is usually of low capacity, thus leading to inaccurate descriptions of target concepts. By embedding the input data into a high capacity set of linear hypotheses and then constructing features in a greedy manner, the proposed method can construct a high capacity feature space. The experimental results also demonstrate the effectiveness of the proposed approach. Despite of its significance, I think the paper can be improved in the following aspects. First, the flow of this paper should be improved. Although English grammar is ok, this paper is not very easy to follow. I understand that the authors need to meet the page limits, but it would be better if the authors can include several sentences in the introduction part to summarize the proposed method, which serve as a clue for the following paragraphs. Second, the effectiveness of the proposed method has been clearly demonstrated by the experiments. Is it possible for the authors to evaluate the classification accuracy for the proposed approach? Therefore, my opinion for this paper is borderline.

Confidence in this Review

2-Confident (read it all; understood it all reasonably well)


Reviewer 4

Summary

The paper proposes an approach that enjoys existing algorithms' scalability and allows high capacities on input features. Authors also exploit tools in functional analysis to support the proposed algorithm. The greedy algorithm performs well in real-world data sets. In summary, the approach is fascinating in the both theory and application.

Qualitative Assessment

Overall, the paper is well-written and well-organized. The mathematical theory is displayed appropriately from the basic knowledge to establishing the convergence rate. The experiments prove the competing performance of the proposed algorithm. To my knowledge, this is a good paper worthy of recommendation.

Confidence in this Review

1-Less confident (might not have understood significant parts)